# Key virulence factors responsible for differences in pathogenicity between clinically proven live-attenuated Japanese encephalitis vaccine SA$_{14}$-14-2 and its pre-attenuated highly virulent parent SA$_{14}$

**Byung-Hak Song**[☯]**, Sang-Im Yun**[☯]**, Joseph L. Goldhardt, Jiyoun Kim, Young-Min Lee**[ID] *

Department of Animal, Dairy, and Veterinary Sciences, College of Agriculture and Applied Sciences, Utah State University, Logan, Utah, United States of America

☯ These authors contributed equally to this work.

* youngmin.lee@usu.edu

**Data Availability Statement:** All relevant data are within the manuscript and its Supporting

## Abstract

Japanese encephalitis virus (JEV), a neuroinvasive and neurovirulent orthoflavivirus, can be prevented in humans with the SA$_{14}$-14-2 vaccine, a live-attenuated version derived from the wild-type SA$_{14}$ strain. To determine the viral factors responsible for the differences in pathogenicity between SA$_{14}$ and SA$_{14}$-14-2, we initially established a reverse genetics system that includes a pair of full-length infectious cDNAs for both strains. Using this cDNA pair, we then systematically exchanged genomic regions between SA$_{14}$ and SA$_{14}$-14-2 to generate 20 chimeric viruses and evaluated their replication capability in cell culture and their pathogenic potential in mice. Our findings revealed the following: (*i*) The single envelope (E) protein of SA$_{14}$-14-2, which contains nine mutations (eight in the ectodomain and one in the stem region), is both necessary and sufficient to render SA$_{14}$ non-neuroinvasive and non-neurovirulent. (*ii*) Conversely, the E protein of SA$_{14}$ alone is necessary for SA$_{14}$-14-2 to become highly neurovirulent, but it is not sufficient to make it highly neuroinvasive. (*iii*) The limited neuroinvasiveness of an SA$_{14}$-14-2 derivative that contains the E gene of SA$_{14}$ significantly increases (approaching that of the wild-type strain) when two viral non-structural proteins are replaced by their counterparts from SA$_{14}$: (*a*) NS1/1', which has four mutations on the external surface of the core β-ladder domain; and (*b*) NS2A, which has two mutations in the N-terminal region, including two non-transmembrane α-helices. In line with their roles in viral pathogenicity, the E, NS1/1', and NS2A genes all contribute to the enhanced spread of the virus in cell culture. Collectively, our data reveal for the first time that the E protein of JEV has a dual function: It is the master regulator of viral neurovirulence and also the primary initiator of viral neuroinvasion. After the initial E-mediated neuroinvasion, the NS1/1' and NS2A proteins act as secondary promoters, further amplifying viral neuroinvasiveness.

information files. The values used to build graphs showing dose-response survival curves (Figs 5, 7, 9, and S2) and viral loads in spleens and brains (Fig 10) are available in Figshare with the following DOIs: Fig 5: https://doi.org/10.6084/m9.figshare.27987743.v1 Fig 7: https://doi.org/10.6084/m9.figshare.27987749.v1 Fig 9: https://doi.org/10.6084/m9.figshare.27987746.v1 Fig 10: https://doi.org/10.6084/m9.figshare.27987740.v1 Fig S2: https://doi.org/10.6084/m9.figshare.27987752.v1.

**Funding:** This research was supported by grants from the Utah Science Technology and Research Initiative (USTAR A34637 and A36815 to YML), as well as the Utah Agricultural Experiment Station (UAES UTA01345 and UTA01655 to YML). This work was approved by the UAES as journal paper number 9303. The funders had no role in study design, data collection and analysis, decision to publish, or preparation of the manuscript.

**Competing interests:** The authors have declared that no competing interests exist.

## Author summary

Although JEV is a vaccine-preventable pathogen, it has remained the primary cause of severe encephalitis in the Asia-Pacific region, the world's most densely populated area, for almost a century since its discovery. Recently, the first detection of JEV in Europe and Africa has raised alarms regarding global public health. Live-attenuated vaccine SA$_{14}$-14-2, derived from the highly virulent JEV strain SA$_{14}$, is the most commonly used vaccine in countries where JEV is endemic. Despite the critical role of this vaccine, little is known about the viral factors that render SA$_{14}$-14-2 fully attenuated and unable to cause Japanese encephalitis, and conversely, the viral factors that make SA$_{14}$ highly virulent and capable of causing the disease. To address these two related yet unanswered questions, we have now used cell culture and mouse infection models pertinent to the attenuation of SA$_{14}$ into SA$_{14}$-14-2 to conduct the first comprehensive, unbiased, reciprocal, and systematic genetic analysis of their variants as well as structural mapping. Our aim was to identify the key virulence factors that account for the striking differences in viral pathogenicity (specifically neuroinvasiveness and neurovirulence) between SA$_{14}$ and SA$_{14}$-14-2. The results of this study establish a robust and extendable framework that elucidates the intricate viral genetics governing JEV pathogenicity. In addition, this research offers new potential molecular targets, thereby laying the groundwork for the design of effective antiviral strategies against JEV infection. Furthermore, our discoveries may apply to other medically important neurotropic orthoflaviviruses, thus broadening our knowledge of viral pathogenicity and facilitating the development of broad-spectrum antiviral approaches for these related pathogens.

## Introduction

Japanese encephalitis virus (JEV) is the prototype species of the JEV serogroup within the genus *Orthoflavivirus* of the family *Flaviviridae* [1]. This serogroup includes several other antigenically related orthoflaviviruses known to cause encephalitis, such as West Nile virus (WNV), St. Louis encephalitis virus (SLEV), Murray Valley encephalitis virus (MVEV), and Usutu virus (USUV) [2–4]. Other orthoflaviviruses, such as Zika virus (ZIKV), dengue virus (DENV), yellow fever virus (YFV), Powassan virus, and tick-borne encephalitis virus are also associated with neurological complications [5–7]. As an arbovirus, JEV is transmitted through an enzootic cycle between culicine mosquito vectors (e.g., *Culex tritaeniorhynchus*) and susceptible vertebrate hosts (e.g., suids and avians), with occasional epizootic spillovers to humans [8–11]. In humans, JEV infection can lead to Japanese encephalitis (JE) [12–14], an inflammatory disease of the central nervous system (CNS) with a case-fatality rate of about 15–25%. Of the survivors, up to 50% suffer from long-term neurological sequelae, including paralysis, memory loss, language problems, and behavioral and emotional disorders [15]. JE is the most prevalent form of viral encephalitis in the Asia-Pacific region [16–19], affecting populous countries like China, India, Indonesia, Pakistan, and Bangladesh, which collectively represent more than 50% of the global population [20]. Beyond the Asia-Pacific, JEV infections in birds and mosquitoes have been detected in Italy since 1995 [21–23], and a case of asymptomatic JEV infection in humans was reported in Angola in 2016 [24]. Currently, JEV is considered an emerging threat to the US, although the country is not yet affected by the pathogen [25]. For JEV control, no antiviral drugs have been developed [26], but killed and live vaccines are available for human use [27,28]. Despite the availability of multiple vaccines, the annual number of

new JE cases in Asia is still estimated to be over 100,000 [29]. Therefore, although JEV is vaccine-preventable, it remains a clinically important pathogen that poses a significant public health threat in increasingly larger parts of the world [30,31], underscoring the urgent need for therapeutic development.

JEV is an enveloped virus with an approximately 11-kb capped, but non-polyadenylated, positive-strand RNA genome. This genome contains a long open reading frame (ORF) flanked by short non-coding regions (NCRs). The ORF encodes a polyprotein precursor comprising three structural proteins (the capsid [C], pre-membrane [prM], and envelope [E]) and seven nonstructural proteins (NS1, NS2A, NS2B, NS3, NS4A, NS4B, and NS5) [32]. The JEV virion houses an inner nucleocapsid formed by 60 C:C dimers bound to the RNA genome [33,34], which is encased in a lipid bilayer with 90 (M:E)$_2$ dimers arranged in a herringbone pattern on its outermost surface [35,36]. The initial step of JEV replication involves the virus entering the host cell, a process mediated by the viral E protein. This protein directs the virus into the receptor-mediated endocytic pathway and facilitates the low pH-dependent fusion of the viral and endosomal membranes, resulting in the release of the RNA genome into the cytoplasm [37–40]. Subsequently, the genomic RNA is translated into a polyprotein, which is then cleaved into the 10 proteins mentioned earlier. This proteolytic cleavage is performed by the viral serine protease located in the N-terminal region of NS3, which requires its cofactor NS2B, and by at least two cellular proteases [41]. During translation, a programmed -1 ribosomal frameshift (-1 PRF) event occurs at codons 8–9 of NS2A [42], leading to the expression of NS1', an NS1 variant with additional 52 amino acids at the C-terminus [43,44]. This event is facilitated by the -1 PRF signal, which includes a slippery heptanucleotide motif positioned upstream of a pseudoknot RNA structure [42]. Next, the genomic RNA is replicated in a specialized organelle derived from the endoplasmic reticulum (ER), built by viral nonstructural proteins [45–48]. The RNA replication is catalyzed by the viral NS5 protein [49], which possesses methyltransferase and guanylyltransferase activity in its N-terminal region and RNA-dependent RNA polymerase activity in its C-terminal region, together with the NS3 protein [50], which has helicase, nucleoside triphosphatase, and RNA triphosphatase activities within its C-terminal region [51, 52]. Following RNA replication, a nascent nucleocapsid buds into the ER lumen [53,54] to form the immature virion, which features 60 (prM:E)$_3$ trimers protruding from the viral membrane [55,56]. This immature virion is transported to the cell surface via the secretory pathway. At the trans-Golgi network, the prM protein undergoes further processing into the M protein by furin [57], transforming the immature virion to a mature one [58,59]. Thus, a model of JEV replication at the cellular level has been relatively well established [60]. However, the mechanism by which JEV causes lethal JE at the organismal level is still not fully understood.

JEV is primarily transmitted to humans through the bite of an infected mosquito. During feeding, the mosquito injects virus-laden saliva into the human host's skin [61]. In the dermis, JEV is thought to infect a subset of mononuclear phagocytes [62–69]. Both the infected immune cells and cell-free virions are believed to migrate from the skin through the vasculature to peripheral lymphoid organs (e.g., lymph nodes and spleen), where the virus may replicate in mononuclear leukocytes [70–73]. This replication can lead to transient viremia [74,75], potentially resulting in a systemic infection that distributes the virus to various organs, including the brain [76]. The pathogenicity of JEV, like all neurotropic viruses, is defined by two intrinsic properties: neuroinvasiveness and neurovirulence [77,78]. Neuroinvasiveness is the ability of JEV to enter the brain, likely by crossing the blood-brain barrier (BBB), a thin layer of brain microvascular endothelial cells (BMECs) [79]. Neuroinvasion may occur via one or more of the following mechanisms [14,80,81]: transcellular transport of virions across the interior of a BMEC, paracellular movement of virions through disrupted tight junctions between

BMECs, or transendothelial migration of infected leukocytes. Neurovirulence, on the other hand, is the ability of JEV to cause pathology in the brain [14,82,83]. JEV initiates neuropathology by replicating in a subset of neurons, leading to neuronal death and subsequent activation of glial cells [84–89]. These activated glia release inflammatory mediators (e.g., cytokines, chemokines, proteases, and growth factors) that contribute to neuropathologic events [90], including bystander neuronal death [91–95], BBB disruption [96–106], and infiltration of inflammatory cells into the brain [107–109], all of which exacerbate neurodegeneration and neuroinflammation. Thus, while a basic framework describing how JEV causes lethal JE has been established, the specific viral determinants of JEV pathogenicity, specifically neuroinvasiveness and neurovirulence, and their underlying mechanisms remain largely unknown.

JE is a devastating human disease, yet it can be prevented by vaccination with the live-attenuated JE vaccine SA$_{14}$-14-2, which is derived from the highly virulent wild-type JEV SA$_{14}$ [110,111]. First approved in China in 1988, SA$_{14}$-14-2 has since become the predominant vaccine in JEV-endemic countries, such as Cambodia, India, Laos, Myanmar, Nepal, South Korea, Sri Lanka, and Thailand [27]. Despite its extensive global use, the reasons for the full attenuation of SA$_{14}$-14-2, as opposed to the high pathogenicity of SA$_{14}$, remain elusive. In our study, we employed a unique, innovative, unbiased, systematic, and reciprocal genetic approach, combined with structural mapping, to identify the viral virulence factors influencing the neuroinvasiveness and neurovirulence of SA$_{14}$ and SA$_{14}$-14-2. These factors were extensively characterized in both cell culture and mouse infection model systems, which are relevant to the biology of SA$_{14}$ and SA$_{14}$-14-2. Our results provide new detailed insights into the pathogenicity of JEV and represent a significant advancement in understanding the pathogenicity of many other clinically important neurotropic orthoflaviviruses, such as WNV, SLEV, MVEV, USUV, and ZIKV. In addition, our findings may have significant implications for the rational design of live-attenuated vaccines against these emerging and re-emerging pathogens.

## Results

### Construction of two full-length infectious cDNA clones for SA$_{14}$ and SA$_{14}$-14-2, a pair of isogenic but pathogenetically distinct JEV strains

The live-attenuated vaccine strain SA$_{14}$-14-2 was developed after >100 passages of the wild-type JEV strain SA$_{14}$ in hamster kidney cells *in vitro* and mice *in vivo* [110]. The genomes of both SA$_{14}$ and SA$_{14}$-14-2 have been sequenced by multiple research groups. However, discrepancies exist in their reported sequences, with the difference ranging from 47 to 64 nucleotides [111–117], likely related to differences in the passage history of the viruses sequenced [117–120]. Consequently, we independently determined the complete genome sequences of the two strains [116,121], using the SA$_{14}$ virus obtained from a national resource repository and the SA$_{14}$-14-2 virus retrieved from a commercial vaccine vial. Our sequence comparison identified 57 nucleotide differences scattered throughout the genome between SA$_{14}$ and SA$_{14}$-14-2: one in the 5'NCR, 55 in the ORF, and one in the 3'NCR (Fig 1). Within the ORF, 25 of these changes are missense mutations (Fig 1A), affecting proteins C (L$^{66}$S), prM (G$^{21}$D), E (L$^{107}$F, E$^{138}$K, I$^{176}$V, T$^{177}$A, E$^{244}$G, Q$^{264}$H, K$^{279}$M, A$^{315}$V, and K$^{439}$R), NS1 (G$^{235}$D, G$^{292}$S, R$^{339}$M, and D$^{351}$H), NS2A (N$^2$K and A$^{40}$V), NS2B (E$^{63}$D and D$^{65}$G), NS3 (M$^{59}$V and A$^{105}$G), NS4B (I$^{106}$V), and NS5 (H$^{386}$Y, Q$^{639}$H, and V$^{671}$A). The remaining 30 changes are silent mutations, including one at codon 22 in NS2A (G$^{3599}$→A), which is predicted to disrupt a conserved pseudoknot RNA structure (Fig 1B). This disruption has been shown to eliminate the -1 PRF signal necessary for the synthesis of NS1' in SA$_{14}$-infected cells [42,43], thereby abolishing its expression in SA$_{14}$-14-2-infected cells [44,116].

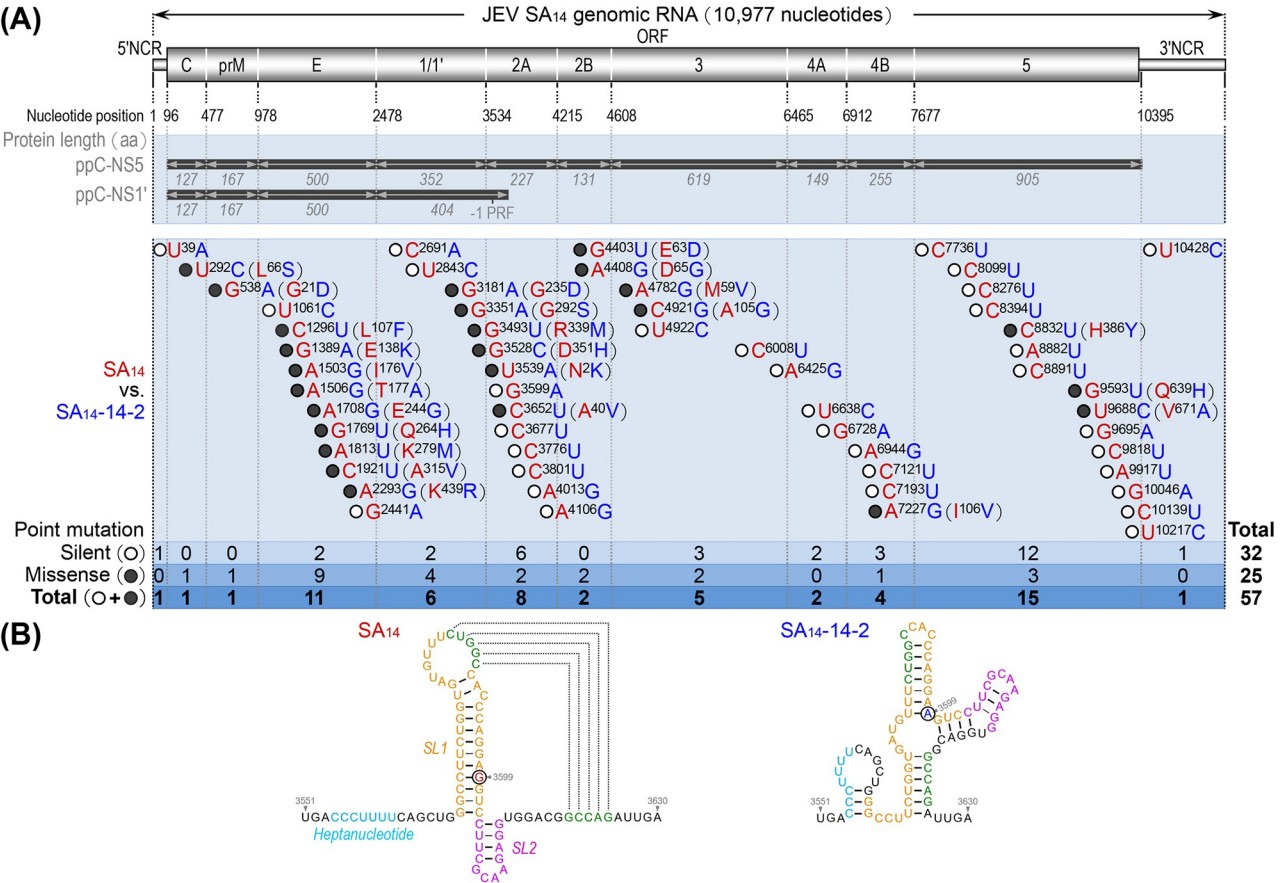

**Fig 1. Comparison of the complete genome sequences of SA14 and SA14-14-2 strains.** (**A**) Variations in the nucleotide and amino acid sequences between the genomes of SA14 and SA14-14-2. The top panel shows a schematic diagram of the SA14 genomic RNA, including a 5' noncoding region (5'NCR), a single open reading frame (ORF), and a 3'NCR. In the ORF, a programmed -1 ribosomal frameshift (-1 PRF) takes place at codons 8–9 of NS2A, leading to the translation of two polyprotein precursors (ppC-NS5 and ppC-NS1'). The two polyproteins undergo proteolytic processing to yield 10 functional proteins: three structural (C, prM, and E) and seven nonstructural (NS1/1', 2A, 2B, 3, 4A, 4B, and 5) proteins. The nucleotide position indicated below the viral genome marks the starting point of each non-coding and protein-coding region. The protein length indicated below the two polyprotein precursors represents the number of amino acids that constitute the corresponding protein. The bottom panel summarizes the 57 nucleotide differences between the genomes of SA14 (shown in red, with the GenBank accession number: KU323483) and SA14-14-2 (shown in blue, with the GenBank accession number: JN604986). This list comprises 32 silent mutations (open circle) and 25 missense mutations (solid circle), with the respective amino acid substitutions indicated in parentheses. The nucleotide positions are determined based on the sequence of the SA14 genomic RNA, and the amino acid positions are determined based on the sequence of the corresponding protein. (**B**) Impact of the silent mutation, $G^{3599}A$, present in the SA14-14-2 genomic RNA on the formation of the RNA pseudoknot predicted to form at nucleotide positions 3551–3630 in the SA14 genomic RNA. This region contains a conserved heptanucleotide sequence (sky blue), two stem-loops (SL1, orange and SL2, pink), and five key base pairs (green) required for pseudoknot formation.

Utilizing our viral genome sequence data, we aimed to develop a reverse genetics system to generate reciprocal chimeric viruses between SA14 and SA14-14-2, by creating their full-length infectious cDNA clones. Previously, we constructed such a clone for SA14-14-2, designated pBac/SA14-14-2, by using a bacterial artificial chromosome (BAC) [121]. Using the same BAC cloning strategy, we obtained a full-length infectious cDNA clone for SA14, designated pBac/SA14 (Fig 2). For both clones, we first synthesized four overlapping cDNA fragments that represent the entire viral RNA genome by RT-PCR. We then assembled these fragments into a full-length cDNA by sequentially joining them at three natural restriction sites (*Bsr*GI, *Bam*HI, and *Ava*I) found in both viral genomes. The resulting full-length cDNA is flanked by a strong SP6 phage promoter at the 5' end and a unique *Xba*I restriction site at the 3'-end. This

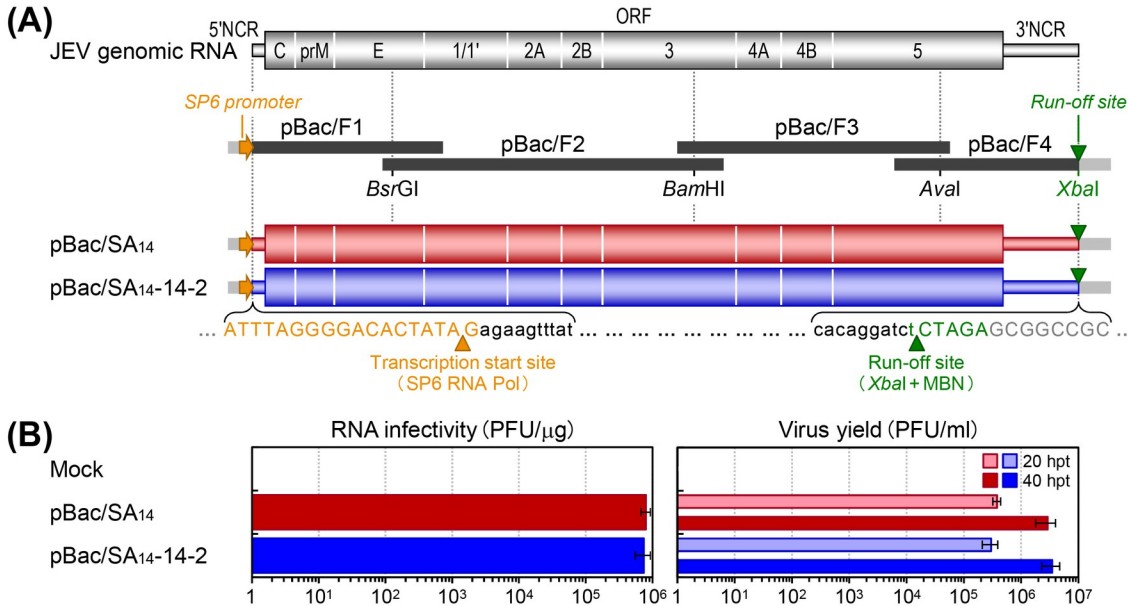

**Fig 2. Generation of two full-length infectious cDNA clones for SA₁₄ and SA₁₄-14-2 strains.** (**A**) Cloning strategy used to assemble two full-length cDNAs, each representing the genomic RNA of SA₁₄ and SA₁₄-14-2. Both viral strains have the same genomic RNA structure, featuring a long open reading frame (ORF) located between the 5' and 3' noncoding regions (NCRs). For each viral strain, the genomic RNA was converted into four overlapping cDNAs, each of which was then subcloned into a bacterial artificial chromosome vector, creating pBac/F1 to pBac/F4. The pBac/F1 subclone was further engineered to introduce the SP6 promoter sequence just before the viral 5'NCR, and the pBac/F4 subclone was modified to incorporate a unique *Xba*I run-off site right after the viral 3'NCR. These four subclones were used to assemble a full-length cDNA using three natural restriction sites (*Bsr*GI, *Bam*HI, and *Ava*I) found in the genomes of both SA₁₄ and SA₁₄-14-2. This assembly process yielded two full-length cDNA clones for SA₁₄ (pBac/SA₁₄) and SA₁₄-14-2 (pBac/SA₁₄-14-2). An orange arrowhead marks the transcription start site, and a green arrowhead indicates the run-off site. (**B**) Functionality of the full-length cDNA clones of SA₁₄ and SA₁₄-14-2. Each cDNA clone was linearized by digestion with *Xba*I and subsequently treated with mung bean nuclease (MBN). The linearized cDNA was then used as a template for *in vitro* run-off transcription with SP6 RNA polymerase (Pol). The resulting RNA transcripts were transfected into BHK-21 cells. These RNA-transfected cells underwent a 10-fold serial dilution to determine their specific infectivity by infectious center assays (RNA infectivity). Culture supernatants were collected from an equal number of RNA-transfected cells at 20 and 40 h post-transfection (hpt) to measure the virus yield by immunoplaque assays (Virus yield).

configuration allows for the *in vitro* run-off transcription of *Xba*I-linearized and mung bean nuclease-treated cDNA using SP6 RNA polymerase, producing synthetic RNAs with the authentic 5' and 3' ends of the viral genomic RNA (Fig 2A). We assessed the functionality of the two full-length cDNA clones, pBac/SA₁₄ and pBac/SA₁₄-14-2, by measuring the specific infectivity of their RNA transcripts as plaque-forming units (PFU) per µg following transfection into baby hamster kidney BHK-21 cells, a type of cell used for the *in vitro* serial passage of SA₁₄ during the development of SA₁₄-14-2. The results showed that both RNA transcripts were equally infectious, with an average infectivity of 7.4–8.0×10⁵ PFU/µg, rapidly producing a high titer of recombinant viruses reaching 3.0–3.8×10⁵ PFU/ml at 20 h post-transfection and further increasing by approximately one log to 2.9–3.5×10⁶ PFU/ml at 40 h post-transfection (Fig 2B).

## Comparison of viral replication in cell culture and pathogenicity in mice between two molecularly cloned recombinant viruses, rSA₁₄ and rSA₁₄-14-2, rescued from their functional cDNA clones

We examined the replication capabilities of rSA₁₄ and rSA₁₄-14-2, each recovered from their respective full-length infectious cDNA clones, in BHK-21 cells after inoculation at a

multiplicity of infection (MOI) of 1 PFU/cell. For comparison, we included the two original viruses used for cDNA construction, SA$_{14}$ and SA$_{14}$-14-2. Our findings indicated that, like the original SA$_{14}$ and SA$_{14}$-14-2 pair, rSA$_{14}$ replicated slightly more effectively than did rSA$_{14}$-14-2, as demonstrated by higher levels of viral genomic RNA (Fig 3A) and proteins (Fig 3B) accumulated during the first 24 h after infection. This accumulation of viral genomic RNA and proteins was consistent with the kinetics of viral growth, which we monitored over a period of 3 days following infection (Fig 3C), and with the size of the viral plaques visualized at 4 days post-infection (Fig 3D). These results indicate that the more efficient RNA replication of rSA$_{14}$ leads to faster viral growth and consequently larger plaque formation. In line with our previous report on the SA$_{14}$ and SA$_{14}$-14-2 pair [116], immunoblot analyses with a panel of 15 JEV antigen-specific rabbit antisera showed no significant differences in the viral protein expression profiles between rSA$_{14}$-infected and rSA$_{14}$-14-2-infected cells, with the exception of the -1 PRF product NS1', which was present in rSA$_{14}$-infected cells but absent from rSA$_{14}$-14-2-infected

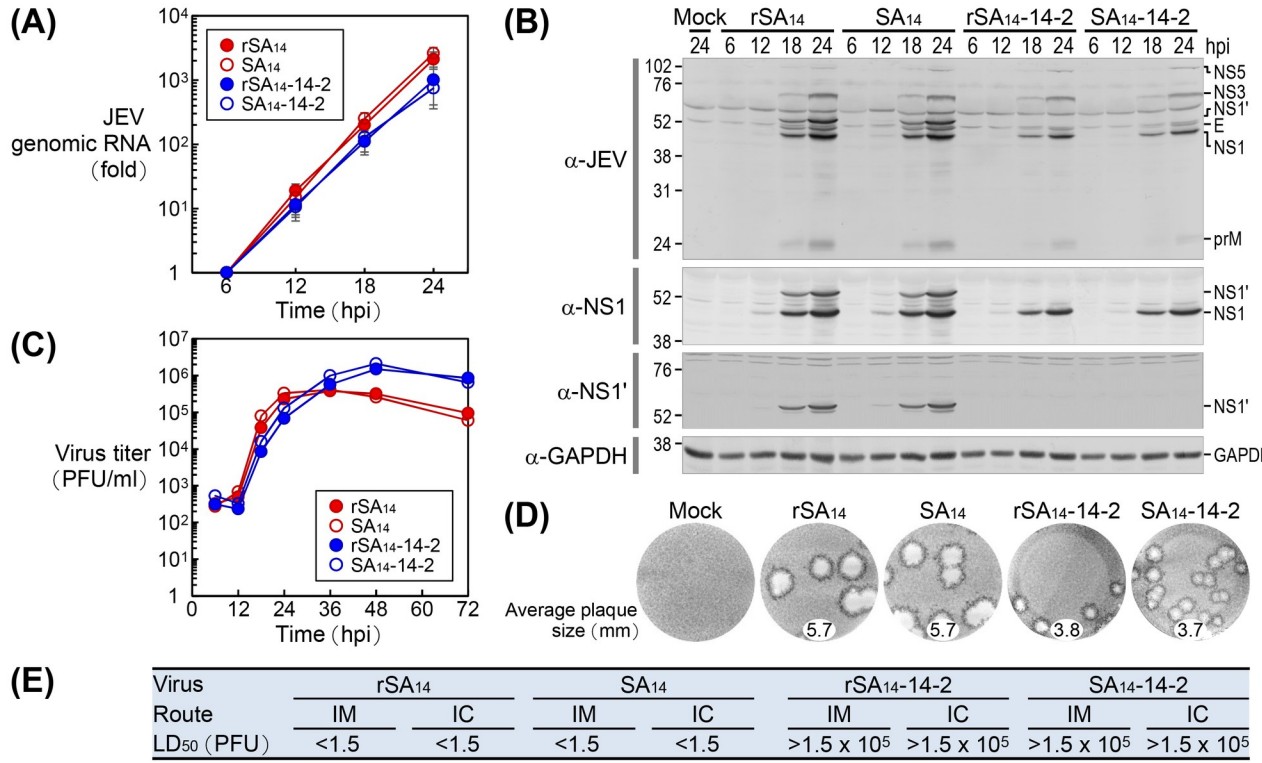

Fig 3. Virological properties of two cDNA-derived recombinant viruses, rSA$_{14}$ and rSA$_{14}$-14-2, *in vitro* and *in vivo*. (A-D) Viral replication in cell culture. BHK-21 cells were either mock-infected or infected at an MOI of 1 with rSA$_{14}$ or rSA$_{14}$-14-2 (each recovered from its respective functional cDNA clone), or with SA$_{14}$ or SA$_{14}$-14-2 (each used for its respective cDNA construction). (A) Viral RNA replication. At 6, 12, 18, and 24 h post-infection (hpi), total cellular RNAs were examined using RT-qPCR with a JEV NS3-specific TaqMan probe. The accumulation levels of viral genomic RNA are expressed as fold changes, relative to the baseline at 6 hpi. (B) Viral protein production. At 6, 12, 18, and 24 hpi, total cell lysates were analyzed using immunoblotting with a mouse hyperimmune antiserum raised against JEV or a rabbit antiserum specific for JEV NS1 or NS1' to determine the accumulation levels of viral proteins. GAPDH was used as an internal control. Molecular weight markers in kDa are indicated on the left, and viral proteins are labeled on the right. (C) Viral growth analysis. At 6, 12, 18, 24, 36, 48, and 72 hpi, culture supernatants were collected to assess the amounts of infectious virus particles released from infected cells by conducting virus titration using immunoplaque assays with a rabbit antiserum specific for JEV NS3. (D) Viral plaque morphology. After 4 days of incubation under a semi-solid overlay, cell monolayers were immunostained with a rabbit antiserum specific for JEV NS3 to visualize viral plaques. The images of representative plaques are provided, along with their average dimensions. (E) Viral pathogenicity in mice. Groups of CD-1 mice (n = 5 per group) were either mock-infected or infected intramuscularly (IM) or intracerebrally (IC) with 10-fold serial dilutions of rSA$_{14}$, SA$_{14}$, rSA$_{14}$-14-2, or SA$_{14}$-14-2, at a dose ranging from 1.5 to $1.5 \times 10^5$ PFU/mouse. The 50% lethal doses (LD$_{50}$s) were determined based on the resulting dose-response survival curves.

cells (Fig 3B). Therefore, our data show that the two cDNA-derived recombinant viruses, rSA$_{14}$ and rSA$_{14}$-14-2, replicate in BHK-21 cells in a manner similar to their respective original viruses used for cDNA construction, with rSA$_{14}$ exhibiting a slight but noticeable advantage in viral replication over rSA$_{14}$-14-2.

Next, we evaluated the pathogenic potential of rSA$_{14}$ and rSA$_{14}$-14-2 in CD-1 mice, a well-characterized animal model for lethal JE [116,121], by creating dose-response survival curves. These survival curves indicated the 50% lethal doses (LD$_{50}$s) and the range of survival times (RSTs) after inoculation of groups of mice with serial 10-fold dilutions of each virus ranging from 1.5 to $1.5 \times 10^5$ PFU/mouse, administered via two different routes: intramuscular (IM) for neuroinvasiveness and intracerebral (IC) for neurovirulence. This experiment also included the two original viruses, SA$_{14}$ and SA$_{14}$-14-2, as references for comparison. The results revealed a clear difference in viral pathogenicity: rSA$_{14}$, like SA$_{14}$, was highly neuroinvasive and highly neurovirulent, consistently yielding both IM and IC LD$_{50}$s of <1.5 PFU, with the IM and IC RSTs of 5 to 10 days and 4 to 8 days, respectively. In stark contrast, rSA$_{14}$-14-2, like SA$_{14}$-14-2, was non-neuroinvasive and non-neurovirulent, invariably producing both IM and IC LD$_{50}$s of $>1.5 \times 10^5$ PFU, with no animal mortality detected up to the maximum IM dose and only 0–20% mortality observed at the maximum IC dose (Fig 3E, also see below for a more detailed description of the data). In all deceased mice, the typical clinical signs of JE, such as lethargy, ruffled fur, hunched posture, tremors, and/or hind limb paralysis, progressively developed over the time prior to death, and the virus titers in their brains ranged from $8.4 \times 10^5$ to $1.1 \times 10^7$ PFU/brain. All of the mock-infected mice (negative control) survived with no signs of disease, as expected. Therefore, our data demonstrate that the two cDNA-derived recombinant viruses, rSA$_{14}$ and rSA$_{14}$-14-2, possess virological properties identical to those of their respective original viruses used for cDNA construction in CD-1 mice. Specifically, we found that rSA$_{14}$ is highly virulent, whereas rSA$_{14}$-14-2 is essentially avirulent.

## Determination of three candidate virulence factors contributing to the differences in pathogenicity between rSA$_{14}$ and rSA$_{14}$-14-2

In our initial effort to identify the viral factor(s) responsible for the changes in pathogenicity between SA$_{14}$ and SA$_{14}$-14-2, we utilized their full-length infectious cDNA clones to perform reciprocal swaps of genomic regions between the two strains, resulting in two sets of three chimeric viruses each, for a total of six (Fig 4A): The first set comprised three rSA$_{14}$ derivatives (designated rSA$_{14}$/5'+3'ncr, rSA$_{14}$/c–ns2a, and rSA$_{14}$/e–ns2a), in which the 5' and 3' non-coding, C-to-NS2A coding, and E-to-NS2A coding regions were each replaced with those of rSA$_{14}$-14-2. Conversely, the second set included three rSA$_{14}$-14-2 derivatives (designated rSA$_{14}$-14-2/5'+3'ncr, rSA$_{14}$-14-2/c–ns2a, and rSA$_{14}$-14-2/e–ns2a), in which the same non-coding and coding regions were each replaced with those of rSA$_{14}$. These reciprocal chimeric viruses were designed to probe the relative contributions of the viral NCRs, the approximately N-terminal half of the viral ORF that includes all structural proteins, and the approximately C-terminal half of the viral ORF that includes most nonstructural proteins, to viral replication in cell culture and pathogenicity in mice.

First, the six reciprocal chimeric viruses were analyzed for their ability to replicate in BHK-21 cells after infection at an MOI of 1; for comparison purposes, we also examined their two parental viruses, rSA$_{14}$ and rSA$_{14}$-14-2. We found that the three rSA$_{14}$ derivatives all replicated as efficiently as their parent rSA$_{14}$ and slightly better than the three rSA$_{14}$-14-2 derivatives, which displayed the same replication efficiency as their parent rSA$_{14}$-14-2, as evident from the accumulation levels of viral genomic RNA (Fig 4B) and proteins (Fig 4C) at 20 h post-infection. In line with previous reports [42–44,116], the -1 PRF product NS1' was expressed only in

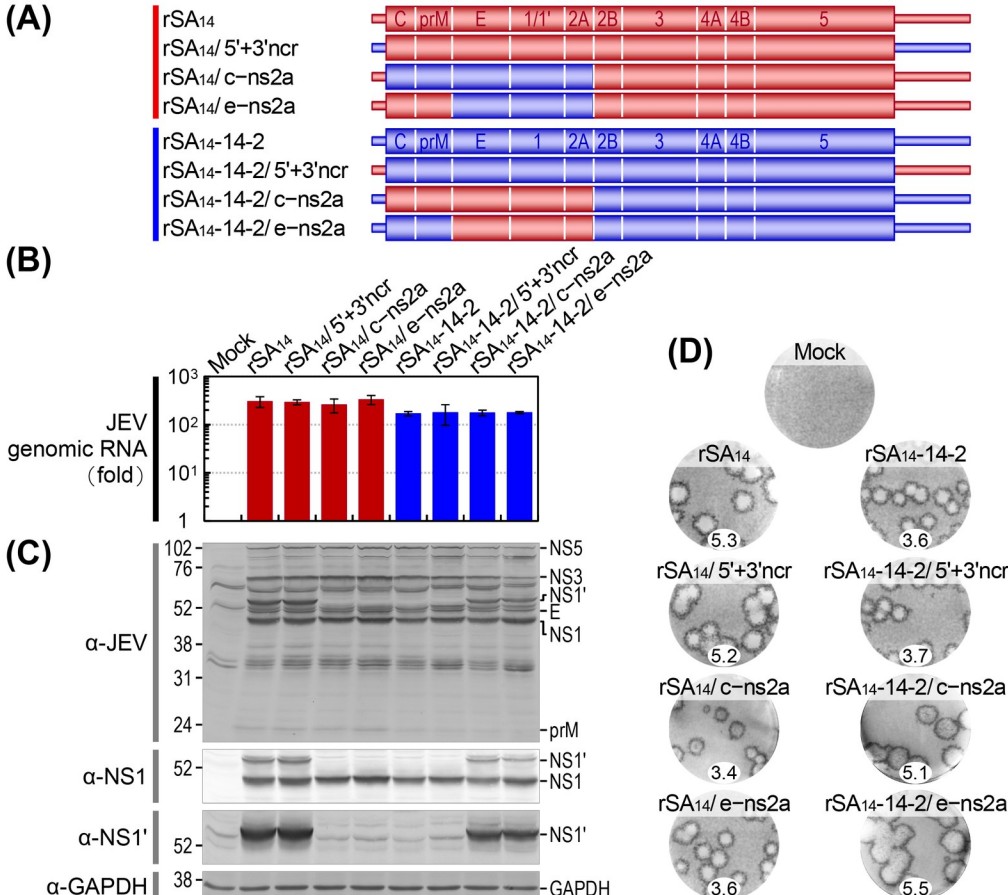

**Fig 4. Initial mapping of the viral genomic region related to the variations in viral replication between rSA₁₄ and rSA₁₄-14-2 in cell culture.** (A) Schematic diagram illustrating the genomes of rSA₁₄, rSA₁₄-14-2, and three pairs of reciprocal chimeric viruses. These chimeric viruses were created by exchanging the 5' and 3' non-coding, C-to-NS2A coding, and E-to-NS2A coding regions between rSA₁₄ and rSA₁₄-14-2. The diagram uses color-coding to distinguish between the exchanged region and the original backbone, with rSA₁₄ shown in red and rSA₁₄-14-2 in blue. (B-D) BHK-21 cells were either mock-infected or infected at an MOI of 1 with each virus as indicated. (B) Viral RNA replication. RT-qPCR was performed on total cellular RNAs using a JEV NS3-specific TaqMan probe to examine the levels of viral genomic RNA accumulation at 20 h post-infection, compared to those at 6 h post-infection. The results are presented as fold changes. (C) Viral protein production. Immunoblotting was conducted on total cell lysates using a mouse anti-JEV hyperimmune antiserum or a rabbit anti-JEV NS1 or anti-JEV NS1' antiserum to analyze the levels of viral protein accumulation at 20 h post-infection. As a loading control, a mouse anti-GAPDH antibody was used to detect GAPDH. Each blot displays molecular weight markers in kDa on the left and viral proteins on the right. (D) Viral plaque morphology. Immunostaining was carried out on cell monolayers using a rabbit anti-JEV NS3 antiserum to detect viral plaques after 4 days of incubation under a semi-solid overlay. Shown are the images of representative plaques with their average dimensions (in mm).

cells infected with rSA₁₄ and its derivative rSA₁₄/5'+3'ncr, as well as two rSA₁₄-14-2 derivatives (rSA₁₄-14-2/c–ns2a and rSA₁₄-14-2/e–ns2a), all of which share the E-to-NS2A coding region of rSA₁₄, with the -1 PRF signal located at the beginning of its NS2A (Fig 4C). We also noted that these three rSA₁₄ or rSA₁₄-14-2 derivatives (i.e., rSA₁₄/5'+3'ncr, rSA₁₄-14-2/c–ns2a, and rSA₁₄-14-2/e–ns2a), which share the E-to-NS2A coding region of rSA₁₄, produced viral plaques as large as those of rSA₁₄, with diameters of 5.1 to 5.5 mm at 4 days post-infection. Conversely, the remaining three rSA₁₄ or rSA₁₄-14-2 derivatives (i.e., rSA₁₄-14-2/5'+3'ncr, rSA₁₄/c–ns2a, and rSA₁₄/e–ns2a), which share the E-to-NS2A coding region of rSA₁₄-14-2, generated

viral plaques as small as those of rSA14-14-2, with diameters of 3.4 to 3.7 mm (Fig 4D). Altogether, the data indicated that while the variation in viral RNA replication efficiency between rSA14 and rSA14-14-2 is associated with the genetic changes that occur in the NS2B-to-NS5 coding region, the difference between the two strains in plaque size, an indication of the efficiency of viral spread to adjacent cells, is linked to the genetic changes that occur in the E-to-NS2A coding region.

Next, the six reciprocal chimeric viruses were assessed for their neuroinvasiveness and neurovirulence in CD-1 mice, as compared to the two parental viruses. This assessment involved generating dose-response survival curves to estimate the LD50s and RSTs after IM and IC inoculations with serial 10-fold dilutions of each virus, ranging from 1.5 to $1.5 \times 10^5$ PFU/mouse (Fig 5). Of the three rSA14 derivatives, rSA14/5'+3'ncr retained the high neuroinvasiveness and high neurovirulence of its parent rSA14, as indicated by both the IM and IC LD50s being <1.5 PFU (Fig 5A). Notably, the IM RST was longer for mice inoculated with rSA14/5'+3'ncr than

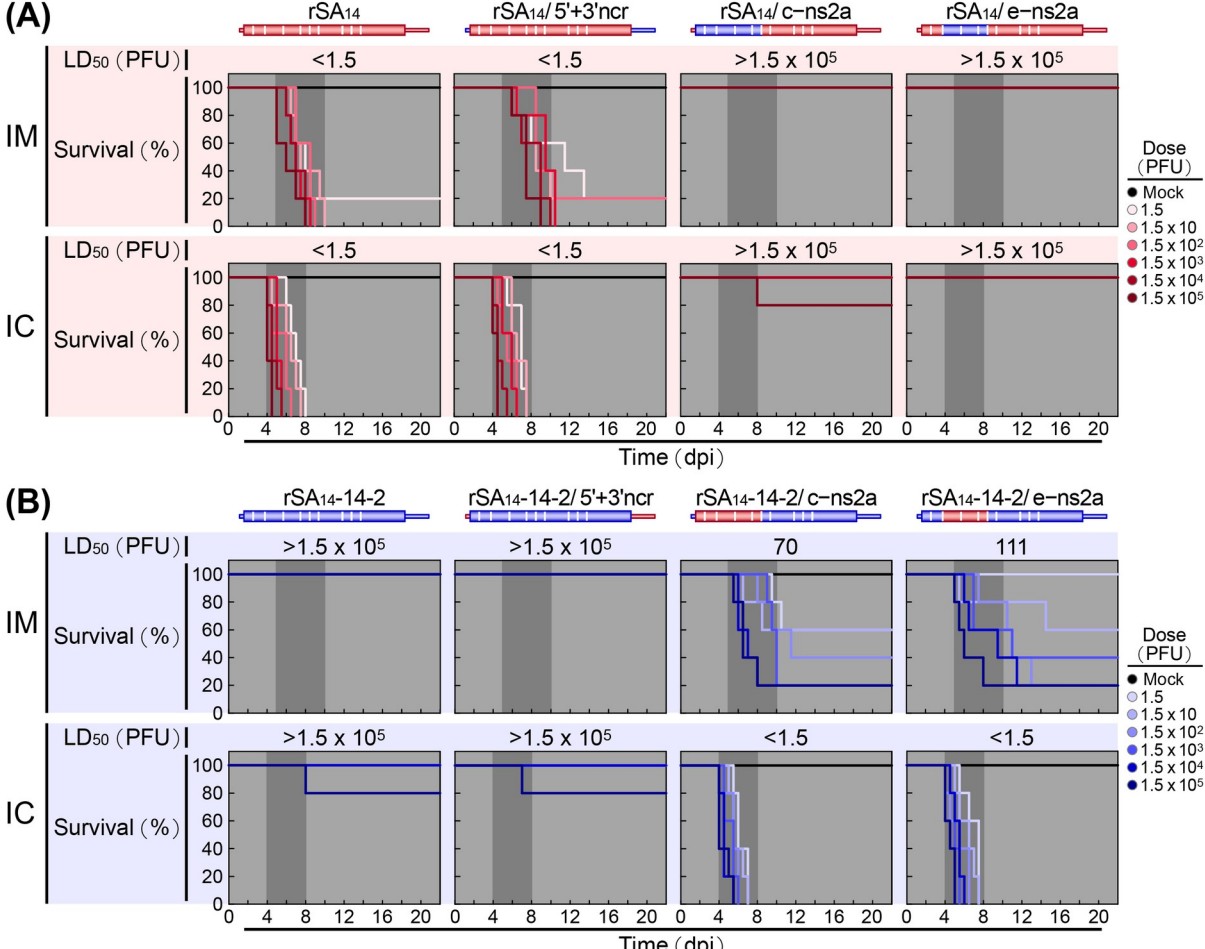

**Fig 5. Initial mapping of the viral genomic region associated with the differences in viral pathogenicity between rSA14 and rSA14-14-2 in mice.** Groups of CD-1 mice (n = 5 per group) were either mock-infected or infected intramuscularly (IM) or intracerebrally (IC) with rSA14, rSA14-14-2 or one of the six reciprocal chimeric viruses described in Fig 4, at a dose ranging from 1.5 to $1.5 \times 10^5$ PFU/mouse. Over a period of 22 days, the mice were monitored twice daily for clinical signs and mortality. The collected data were used to construct dose-response survival curves, which were then used to calculate the 50% lethal doses (LD50s). **Panel A** presents the results of rSA14 and its three derivatives, and **Panel B** displays the results of rSA14-14-2 and its three derivatives. dpi, days post-infection.

for those inoculated with its parent rSA$_{14}$ (6–14 days vs. 5–10 days), whereas the IC RST remained the same for both groups (4–8 days). In contrast, rSA$_{14}$/c−ns2a and rSA$_{14}$/e−ns2a lost neuroinvasiveness and neurovirulence, mirroring the non-neuroinvasive and non-neurovirulent rSA$_{14}$-14-2, with both the IM and IC LD$_{50}$s being $>1.5\times10^5$ PFU; no mortality was observed after IM inoculation up to the maximum dose, and only 0–20% mortality after IC inoculation with the maximum dose (Fig 5A). Conversely, of the three rSA$_{14}$-14-2 derivatives, rSA$_{14}$-14-2/5'+3'ncr remained non-neuroinvasive and non-neurovirulent, like its parent rSA$_{14}$-14-2, with both the IM and IC LD$_{50}$s being $>1.5\times10^5$ PFU; no mortality was detected after the maximum-dose IM inoculation, and only 20% mortality after the maximum-dose IC inoculation (Fig 5B). However, rSA$_{14}$-14-2/c−ns2a and rSA$_{14}$-14-2/e−ns2a became strongly (but not highly) neuroinvasive and highly neurovirulent, with an IM LD$_{50}$ of 70 to 111 PFU and an extended IM RST of 5 to 15 days, and with an IC LD$_{50}$ of $<1.5$ PFU and a typical IC RST of 4 to 8 days (Fig 5B). Collectively, the data indicate that the marked differences in pathogenicity between rSA$_{14}$ and rSA$_{14}$-14-2 are primarily due to genetic changes in the E-to-NS2A coding region, which are capable of converting rSA$_{14}$ into a non-neuroinvasive and non-neurovirulent form, as well as converting rSA$_{14}$-14-2 into a strongly neuroinvasive and highly neurovirulent form. The findings also suggest that while neurovirulence is determined by one or a few viral factors encoded in the E-to-NS2A coding region, neuroinvasiveness is attributed to multiple viral elements, encoded not only primarily in the E-to-NS2A and secondarily in the NS2B-to-NS5 coding regions, which affect the host survival rate, but also in the 5' and 3'NCRs, which influence host survival time.

## Identification of the viral envelope protein E as both the master regulator of neurovirulence and the primary initiator of neuroinvasion, with the viral nonstructural proteins NS1/1' and potentially NS2A acting as secondary promoters of neuroinvasiveness

To identify a specific viral factor(s) among the three candidates (E, NS1/1', and NS2A) within the E-to-NS2A coding region that is crucial for determining viral neuroinvasiveness and neurovirulence, we created two new sets, each comprising three chimeric viruses (six in total; Fig 6A). The first set, designed to be compared with the previously analyzed rSA$_{14}$/e−ns2a and its parent rSA$_{14}$, comprised three rSA$_{14}$ derivatives: (*i*) rSA$_{14}$/e+ns1+prf$^{Neg}$, in which the E and NS1 genes were replaced by their counterparts from rSA$_{14}$-14-2, together with the silent G$^{3599}\rightarrow$A mutation in NS2A, which abolishes the -1 PRF signal necessary for NS1' synthesis; (*ii*) rSA$_{14}$/e, with the E gene replaced by its counterpart from rSA$_{14}$-14-2; and (*iii*) rSA$_{14}$/ns1+prf$^{Neg}$, with the NS1 gene replaced by its counterpart from rSA$_{14}$-14-2, plus the silent G$^{3599}\rightarrow$A mutation disrupting the -1 PRF in NS2A. Conversely, the second set, created for comparison with the previously examined rSA$_{14}$-14-2/e−ns2a and its parent rSA$_{14}$-14-2, comprised three rSA$_{14}$-14-2 derivatives: (*a*) rSA$_{14}$-14-2/e+ns1+prf$^{Pos}$, with the E and NS1 genes replaced by their counterparts from rSA$_{14}$, together with the silent A$^{3599}\rightarrow$G mutation in NS2A, which restores the -1 PRF signal for NS1' synthesis; (*b*) rSA$_{14}$-14-2/e, with the E gene replaced by its counterpart from rSA$_{14}$; and (*c*) rSA$_{14}$-14-2/ns1+prf$^{Pos}$, with the NS1 gene replaced by its counterpart from rSA$_{14}$, plus the silent A$^{3599}\rightarrow$G mutation restoring the -1 PRF in NS2A.

Initially, we examined the replication ability of the six new reciprocal chimeric viruses and their two parental viruses in BHK-21 cells following infection at an MOI of 1. Our results demonstrated that rSA$_{14}$, rSA$_{14}$/e−ns2a, and all three rSA$_{14}$ derivatives not only replicated equally well but also slightly outperformed rSA$_{14}$-14-2, rSA$_{14}$-14-2/e−ns2a, and all three rSA$_{14}$-14-2 derivatives, as evidenced by the accumulation levels of viral genomic RNA (Fig 6B) and

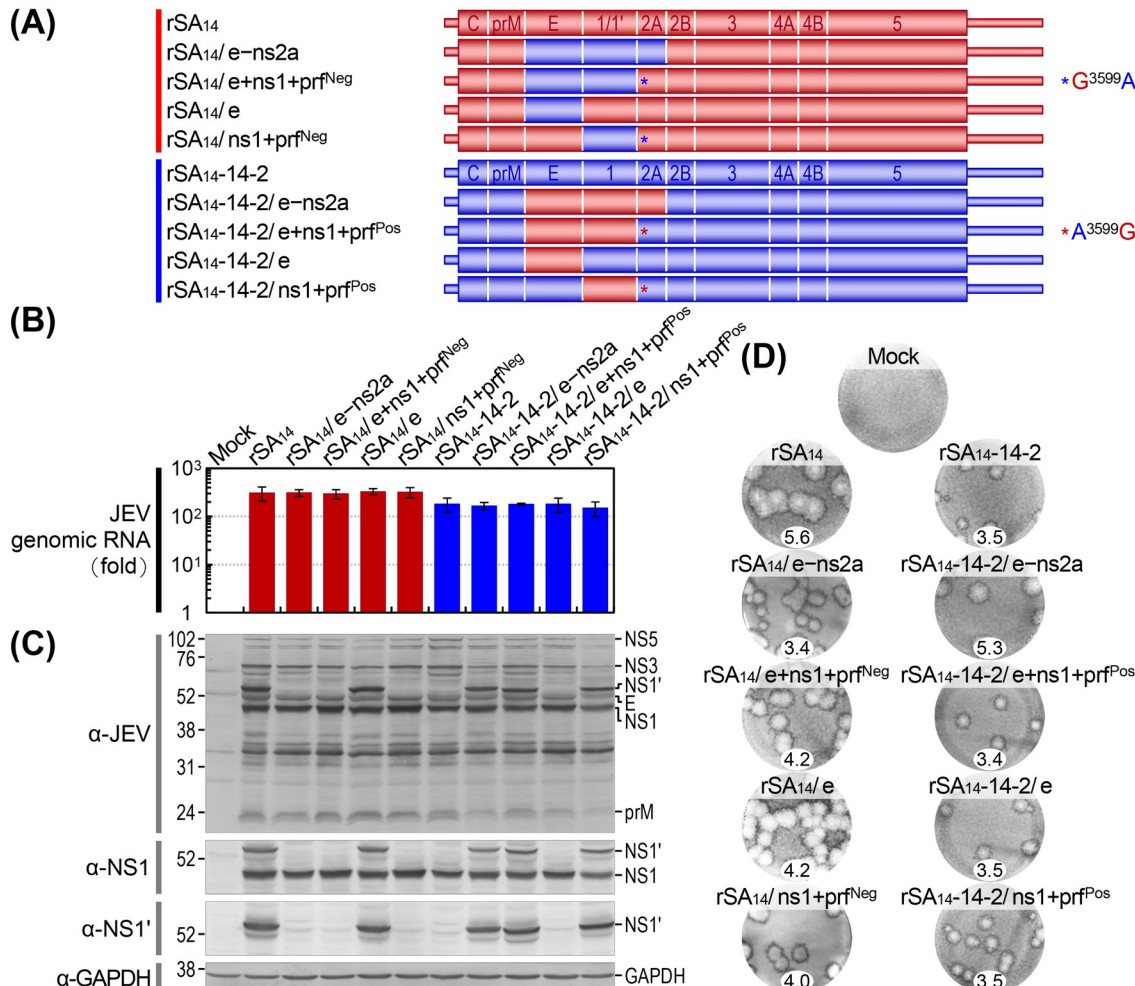

**Fig 6. Identification of the key viral genetic factors contributing to the variations in viral replication between rSA$_{14}$ and rSA$_{14}$-14-2 in cell culture.** (**A**) Schematic diagram presenting the genomes of rSA$_{14}$, rSA$_{14}$-14-2, and four pairs of reciprocal chimeric viruses. These chimeric viruses were generated by swapping the E gene, the NS1 gene with a silent G$^{3599}$A or A$^{3599}$G mutation in NS2A, the E and NS1 genes with the same silent mutation, and the E, NS1, and NS2A genes between rSA$_{14}$ and rSA$_{14}$-14-2. Color-coding differentiates the genes and mutations that were replaced in the original backbone: rSA$_{14}$ is in red, rSA$_{14}$-14-2 in blue. (**B-D**) BHK-21 cells were either mock-infected or infected with each of the specified viruses at an MOI of 1. (**B**) Viral RNA replication. Total cellular RNAs were examined using RT-qPCR with a JEV NS3-specific TaqMan probe to measure the levels of viral genomic RNA accumulation at 20 h post-infection, relative to those at 6 h post-infection. The data are shown as fold changes. (**C**) Viral protein production. Total cell lysates were analyzed using immunoblotting with a mouse anti-JEV hyperimmune antiserum, rabbit anti-JEV NS1, or anti-JEV NS1' antiserum to determine the levels of viral protein accumulation at 20 h post-infection. GAPDH was used as a loading control and detected with mouse anti-GAPDH antibody. Molecular weight markers in kDa are shown on the left of each blot, and viral proteins are indicated on the right. (**D**) Viral plaque morphology. Cell monolayers were stained with a rabbit anti-JEV NS3 antiserum to detect viral plaques after a 4-day incubation period under a semi-solid overlay. Representative plaques are shown along with their average dimensions (mm).

proteins (Fig 6C) over 20 h after infection. As expected, we also observed that a single G$^{3599}$→A or A$^{3599}$→G substitution in the genetic background of rSA$_{14}$ or rSA$_{14}$-14-2 was sufficient to abolish or restore the -1 PRF-mediated expression of NS1', respectively (Fig 6C). Interestingly, at 4 days post-infection, all three rSA$_{14}$ derivatives formed medium-sized plaques (diameter, 4.0 to 4.2 mm) that were smaller than those produced by rSA$_{14}$ and rSA$_{14}$-14-2/e–ns2a (5.3 to 5.6 mm) but larger than those produced by rSA$_{14}$-14-2 and rSA$_{14}$/e–ns2a (3.4 to

3.5 mm) (Fig 6D). On the other hand, all three rSA₁₄-14-2 derivatives formed small plaques (diameter, 3.4 to 3.5 mm), indistinguishable from those produced by rSA₁₄-14-2 and rSA₁₄/e−ns2a (Fig 6D). These results corroborate our earlier findings that the difference in plaque size, indicative of the efficiency of viral spread, between rSA₁₄ and rSA₁₄-14-2 is caused by genetic changes accumulated in the E-to-NS2A coding region. More specifically, our data indicate that a minor reduction in the plaque size of rSA₁₄ from large to medium occurs when the E or NS1/1' gene is replaced, either alone or in combination, with that of rSA₁₄-14-2, but that a major reduction from large to small is only achieved by simultaneously replacing all three genes, E, NS1/1', and NS2A, resulting in a plaque size as small as those of rSA₁₄-14-2. Similarly, our results also indicate that the small plaque size of rSA₁₄-14-2 does not change noticeably when the E or NS1/1' gene is swapped, either individually or in combination, with that of rSA₁₄; however, it increases significantly from small to large when all three genes, E, NS1/1', and NS2A, are simultaneously swapped (see S1 Fig for additional data).

Subsequently, we analyzed the neuroinvasiveness and neurovirulence of our six new reciprocal chimeric viruses in CD-1 mice, in relation to the two parental viruses, by using the $LD_{50}$s and RSTs determined from the dose-response survival curves after IM and IC inoculations with serial 10-fold dilutions of each virus from 1.5 to $1.5 \times 10^5$ PFU/mouse (Fig 7). Our results

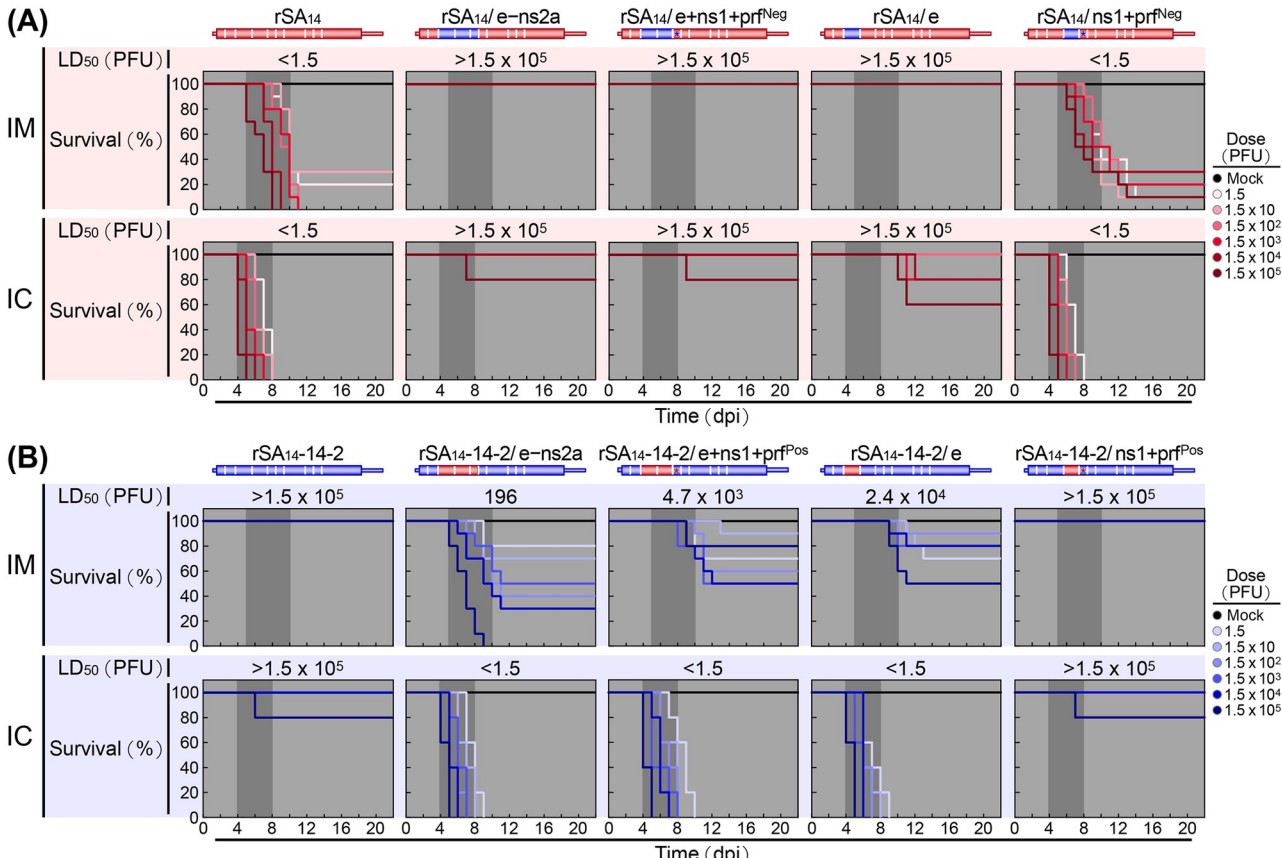

**Fig 7. Identification of the key viral genetic factors responsible for the differences in viral pathogenicity between rSA₁₄ and rSA₁₄-14-2 in mice.**
Groups of CD-1 mice were either mock-infected or infected intramuscularly (IM, n = 10 per group) or intracerebrally (IC, n = 5 per group) with rSA₁₄, rSA₁₄-14-2, or one of the eight reciprocal chimeric viruses detailed in Fig 6, at a dose ranging from 1.5 to $1.5 \times 10^5$ PFU/mouse. Over a period of 22 days, the mice were observed daily for clinical signs and mortality to generate dose-response survival curves. The 50% lethal doses ($LD_{50}$s) were then determined based on these survival curves. (**A**) Outcomes for rSA₁₄ and its four derivatives. (**B**) Outcomes for rSA₁₄-14-2 and its four derivatives. dpi, days post-infection.

showed that among the three rSA$_{14}$ derivatives, rSA$_{14}$/ns1+prf$^{Neg}$ remained highly neuroinvasive and highly neurovirulent, like rSA$_{14}$, with both IM and IC LD$_{50}$s estimated to be <1.5 PFU; yet the IM RST was longer than that of its parent rSA$_{14}$ (6 to 14 days vs. 5 to 11 days), while the IC RST was identical (4 to 8 days) (Fig 7A). In contrast, rSA$_{14}$/e was converted to a non-neuroinvasive and non-neurovirulent form, like rSA$_{14}$-14-2, with both IM and IC LD$_{50}$s estimated to be >1.5×10$^5$ PFU; however, it began to cause lethal encephalitis with even a moderate dose of the chimeric virus administered via the IC route, but never via the IM route (Fig 7A). Specifically, there was a 20% mortality with a dose of either 1.5×10$^3$ or 1.5×10$^4$ PFU/mouse, and a 40% mortality with 1.5×10$^5$ PFU/mouse at 10 to 12 days post-infection (Fig 7A, IC). In comparison, rSA$_{14}$-14-2 typically caused <20% mortality only with 1.5×10$^5$ PFU/mouse at 4 to 8 days post-infection (Fig 7B, IC). Like rSA$_{14}$/e, rSA$_{14}$/e+ns1+prf$^{Neg}$ also appeared in a non-neuroinvasive and non-neurovirulent form, with both the IM and IC LD$_{50}$s estimated to be >1.5×10$^5$ PFU; unlike rSA$_{14}$/e, however, its IM and IC RSTs resembled those of rSA$_{14}$-14-2 and rSA$_{14}$/e−ns2a, resulting in no animal mortality after IM inoculation up to the maximum dose and a 20% mortality only after IC inoculation with the maximum dose (Fig 7A). Thus, the assessment of the three rSA$_{14}$ derivatives has demonstrated that the E gene of rSA$_{14}$-14-2 alone is necessary and sufficient to convert rSA$_{14}$ into a non-neuroinvasive and non-neurovirulent form. It also indicates that the NS1/1' gene of rSA$_{14}$-14-2 alone cannot attenuate the IM and IC LD$_{50}$s of rSA$_{14}$ to a detectable level but can specifically extend the IM RST, but not the IC RST.

In addition, our data demonstrated that among the three rSA$_{14}$-14-2 derivatives, rSA$_{14}$-14-2/ns1+prf$^{Pos}$ persisted in a non-neuroinvasive and non-neurovirulent form that was essentially identical to its parent rSA$_{14}$-14-2, with both the IM and IC LD$_{50}$s estimated to be >1.5×10$^5$ PFU (Fig 7B). On the other hand, rSA$_{14}$-14-2/e became weakly neuroinvasive yet highly neurovirulent, as indicated by an IM LD$_{50}$ of 2.4×10$^4$ PFU, with a longer IM RST than that for rSA$_{14}$ (9 to 13 days vs. 5 to 11 days), and an IC LD$_{50}$ of <1.5 PFU, with nearly the same IC RST as rSA$_{14}$ (4 to 9 days vs. 4 to 8 days) (Fig 7B). Similarly, rSA$_{14}$-14-2/e+ns1+prf$^{Pos}$ appeared in a moderately neuroinvasive but highly neurovirulent form, as indicated by an IM LD$_{50}$ of 4.7×10$^3$ PFU, with a longer IM RST than that of rSA$_{14}$ (8 to 13 days vs. 5 to 11 days), and an IC LD$_{50}$ of <1.5 PFU, with an almost identical IC RST to rSA$_{14}$ (4 to 10 days vs. 4 to 8 days) (Fig 7B). As shown earlier, rSA$_{14}$-14-2/e−ns2a was confirmed to be a strongly neuroinvasive and highly neurovirulent form, as determined by an IM LD$_{50}$ of 196 PFU, with an IM RST of 5 to 11 days, and an IC LD$_{50}$ of <1.5 PFU, with an IC RST of 4 to 9 days (Fig 7B). Hence, our evaluation of these three rSA$_{14}$-14-2 derivatives has shown that: (*i*) the E gene of rSA$_{14}$ alone is necessary and sufficient to convert rSA$_{14}$-14-2 into a highly neurovirulent form; (*ii*) the E gene of rSA$_{14}$ is also still required to render rSA$_{14}$-14-2 capable of initiating neuroinvasion but is not sufficient to convert rSA$_{14}$-14-2 to a highly neuroinvasive form; and (*iii*) the two viral nonstructural protein genes, NS1/1' and NS2A, of rSA$_{14}$ have the ability to promote the weak neuroinvasiveness of rSA$_{14}$-14-2/e and convert it into a strongly neuroinvasive form. In particular, a role for rSA$_{14}$ NS1 in promoting viral neuroinvasiveness was further corroborated in the absence of NS1' expression by a comparative follow-up analysis of rSA$_{14}$-14-2, rSA$_{14}$-14-2/e−ns2a, and two new additional rSA$_{14}$-14-2 derivatives: (*a*) rSA$_{14}$-14-2/e+ns1, which contains the E and NS1 genes replaced with their counterparts from rSA$_{14}$, and (*b*) rSA$_{14}$-14-2/ns1, which contains the NS1 gene replaced with its counterpart from rSA$_{14}$ (S1 and S2 Figs). Thus, our results show that the E gene acts both as the sole determinant controlling neurovirulence and as the key initiator triggering neuroinvasion. After the initial neuroinvasion triggered by the E gene, the NS1/1' and possibly NS2A genes subsequently serve as major promoters enhancing neuroinvasiveness (see below for a full description of the data on the role of NS2A in viral neuroinvasiveness).

## Validation of the viral nonstructural protein NS2A gene as a major promoter of neuroinvasiveness that can enhance neuroinvasion but does not trigger neuroinvasion on its own

To further substantiate the role of rSA$_{14}$ NS2A in enhancing the neuroinvasiveness of rSA$_{14}$-14-2, we generated a final set of six chimeric viruses derived from rSA$_{14}$-14-2 (Fig 8A). The first five chimeric viruses were created by either replacing the NS2A gene with that of rSA$_{14}$,

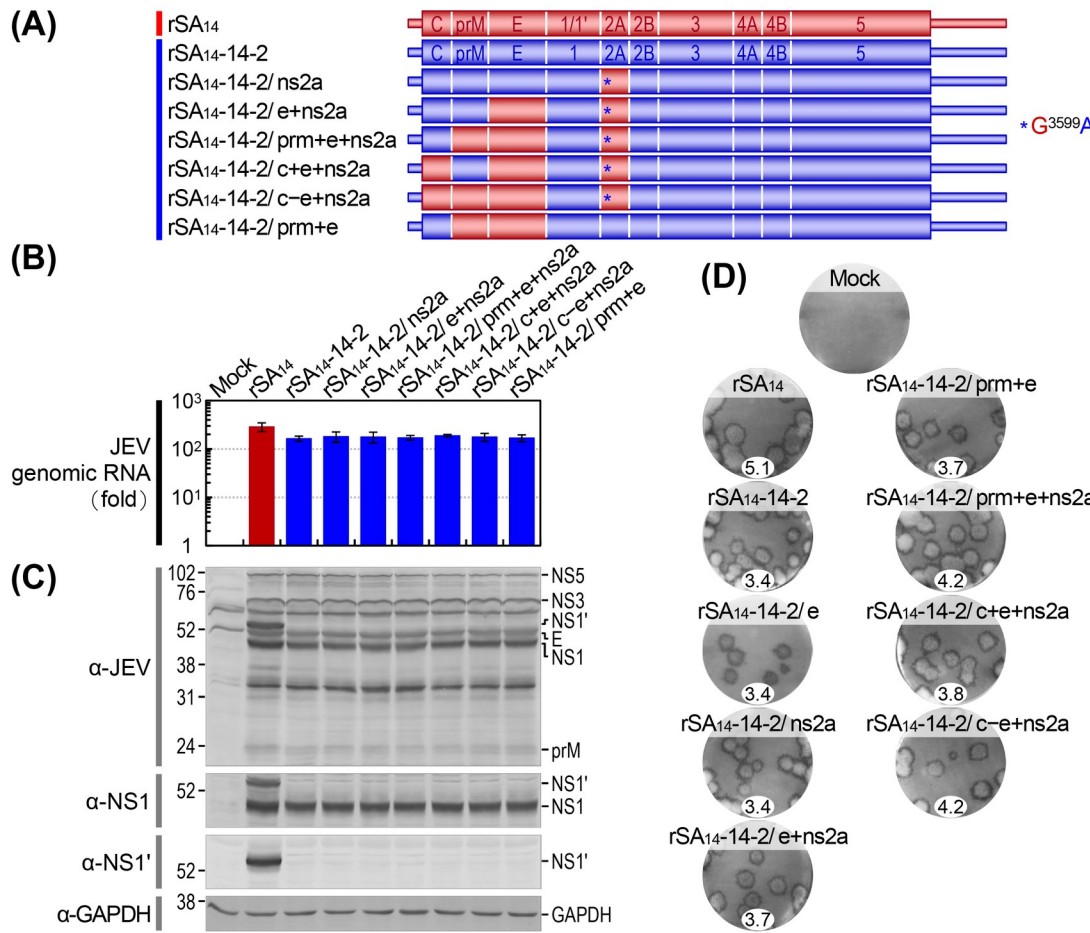

**Fig 8. Validation of the role of the viral NS2A gene in the variations in viral replication between rSA$_{14}$ and rSA$_{14}$-14-2 in cell culture.** (**A**) Schematic diagram depicting the genomes of rSA$_{14}$, rSA$_{14}$-14-2, and six rSA$_{14}$-14-2 derivatives. These derivatives were created by replacing the NS2A gene with that of rSA$_{14}$, while keeping a single A$^{3599}$ nucleotide unchanged in that region, or by combining it with further replacements of the following additional genes with those of rSA$_{14}$: E; prM and E; C and E; and C, prM, and E, or by replacing only the prM and E genes with those of rSA$_{14}$. Color-coding indicates the replaced genes, with rSA$_{14}$ in red and rSA$_{14}$-14-2 in blue. (**B-D**) BHK-21 cells were either mock-infected or infected with each indicated virus at an MOI of 1. (**B**) Viral RNA replication. The levels of viral genomic RNA accumulation at 20 h post-infection, compared to those at 6 h post-infection, were examined by RT-qPCR on total cellular RNAs with a JEV NS3-specific TaqMan probe. The results are expressed as fold changes. (**C**) Viral protein production. The levels of viral protein accumulation at 20 h post-infection were analyzed using immunoblotting on total cell lysates with a mouse anti-JEV hyperimmune antiserum or a rabbit anti-JEV NS1 or anti-JEV NS1' antiserum. As a loading control, a mouse anti-GAPDH antibody was used to detect GAPDH. On each blot, the molecular weight markers in kDa are presented on the left, and the viral proteins are shown on the right. (**D**) Viral plaque morphology. Viral plaques were visualized by immunostaining cell monolayers with a rabbit anti-JEV NS3 antiserum after 4 days of incubation under a semi-solid overlay. The images show representative plaques, along with their average dimensions (mm). Note that the chimeric virus rSA$_{14}$-14-2/e, described in Fig 6, was included in the experiment for comparison. The plaque images were arranged to enhance the readability of the main text.

while leaving a single A$^{3599}$ nucleotide unchanged in that region to disrupt the -1 PRF signal for NS1' expression (designated rSA$_{14}$-14-2/ns2a), or by combining it with an additional replacement in the following genes by those from rSA$_{14}$: E (rSA$_{14}$-14-2/e+ns2a), prM and E (rSA$_{14}$-14-2/prm+e+ns2a), C and E (rSA$_{14}$-14-2/c+e+ns2a), or C, prM, and E (rSA$_{14}$-14-2/c−e+ns2a). The sixth chimeric virus, rSA$_{14}$-14-2/prm+e, was created by replacing the prM and E genes with those of rSA$_{14}$ for purposes of comparison.

We first evaluated the replication capacity of these six chimeric viruses in comparison to rSA$_{14}$ and rSA$_{14}$-14-2 in BHK-21 cells after infection at an MOI of 1. Our findings were consistent with the previously described results, indicating that all six chimeric viruses replicated as effectively as the parental rSA$_{14}$-14-2. Their levels of viral replication were uniformly slightly lower than that of rSA$_{14}$, as evidenced by the levels of viral genomic RNA (Fig 8B) and proteins (Fig 8C) accumulated during the first 20 h after infection. As intended, no expression of NS1' was detected in cells infected with any of the six chimeric viruses (Fig 8C). At 4 days post-infection, we observed that the six chimeric viruses formed plaques of various sizes, ranging from 3.4 to 4.2 mm in diameter, and were grouped into three types (Fig 8D): (*a*) rSA$_{14}$-14-2/ns2a formed small plaques (diameter, 3.4 mm), indistinguishable from those formed by rSA$_{14}$-14-2. Note that rSA$_{14}$-14-2/e, included for comparison in this analysis, also formed small plaques (diameter, 3.4 mm) as shown in Fig 6D. (*b*) rSA$_{14}$-14-2/e+ns2a, rSA$_{14}$-14-2/c+e+ns2a, and rSA$_{14}$-14-2/prm+e all developed small plaques (diameter, 3.7 to 3.8 mm), which were marginally larger than those developed by rSA$_{14}$-14-2 (diameter, 3.4 mm). (*c*) Both rSA$_{14}$-14-2/prm+e+ns2a and rSA$_{14}$-14-2/c−e+ns2a produced medium-sized plaques (diameter, 4.2 mm), larger than those produced by rSA$_{14}$-14-2 (diameter, 3.4 mm) but smaller than those produced by rSA$_{14}$ (diameter, 5.1 mm). These results indicated that the small plaque size of rSA$_{14}$-14-2 remained unchanged when the E or NS2A gene was individually replaced with that of rSA$_{14}$, but it increased, albeit to a limited extent, when both the E and NS2A genes were replaced together (Fig 8D, compare rSA$_{14}$-14-2 vs. rSA$_{14}$-14-2/e vs. rSA$_{14}$-14-2/ns2a vs. rSA$_{14}$-14-2/e+ns2a). The putative role of rSA$_{14}$ NS2A in plaque enlargement of rSA$_{14}$-14-2 was further supported by the results obtained when the prM and E genes were co-replaced with those of rSA$_{14}$ (Fig 8D, compare rSA$_{14}$-14-2/prm+e vs. rSA$_{14}$-14-2/prm+e+ns2a). Our data also suggested that the prM gene of rSA$_{14}$ had a positive effect on plaque enlargement of rSA$_{14}$-14-2 when replaced with the E gene of rSA$_{14}$ (Fig 8D, compare rSA$_{14}$-14-2/e vs. rSA$_{14}$-14-2/prm+e) or co-replaced with the E and NS2A genes of rSA$_{14}$ (Fig 8D, compare rSA$_{14}$-14-2/e+ns2a vs. rSA$_{14}$-14-2/prm+e+ns2a and rSA$_{14}$-14-2/c+e+ns2a vs. rSA$_{14}$-14-2/c−e+ns2a). However, the C gene of rSA$_{14}$ had no effect on plaque enlargement of rSA$_{14}$-14-2, even when either the E and NS2A genes or the prM, E, and NS2A genes were co-replaced with those of rSA$_{14}$ (Fig 8D, compare rSA$_{14}$-14-2/e+ns2a vs. rSA$_{14}$-14-2/c+e+ns2a and rSA$_{14}$-14-2/prm+e+ns2a vs. rSA$_{14}$-14-2/c−e+ns2a).

We then assessed the neuroinvasiveness of our six chimeric viruses in CD-1 mice, using rSA$_{14}$ and rSA$_{14}$-14-2 as references. We also included rSA$_{14}$-14-2/e for comparison in this analysis. This assessment was based on the LD$_{50}$s and RSTs determined from the dose-response survival curves generated by IM inoculation with serial 10-fold dilutions of each virus (Fig 9). Among the six chimeric viruses, rSA$_{14}$-14-2/ns2a was as non-neuroinvasive as its parent rSA$_{14}$-14-2, indicated by an IM LD$_{50}$ of >1.5×10$^5$ PFU, with no animal mortality observed up to the maximum dose administered. Conversely, four chimeric viruses (rSA$_{14}$-14-2/e+ns2a, rSA$_{14}$-14-2/prm+e+ns2a, rSA$_{14}$-14-2/c+e+ns2a, and rSA$_{14}$-14-2/c−e+ns2a), which possess the E and NS2A genes of rSA$_{14}$ either alone or with one or both of the C and prM genes of rSA$_{14}$, were all strongly neuroinvasive, as defined by IM LD$_{50}$s of 225 to 276 PFU and IM RSTs ranging from 5 or 6 days to 16 or 18 days. The remaining chimeric virus, rSA$_{14}$-14-2/prm+e, was weakly neuroinvasive, with an IM LD$_{50}$ of 2.1×10$^4$ PFU and an IM RST of 10 to 13

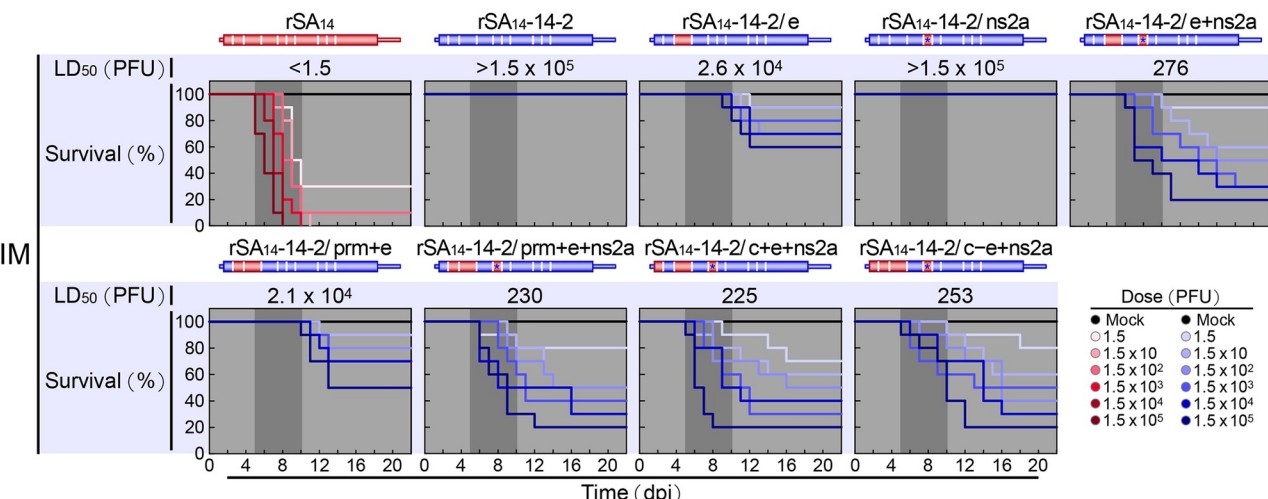

**Fig 9. Validation of the role of the viral NS2A gene in the differences in viral pathogenicity between rSA₁₄ and rSA₁₄-14-2 in mice.** Groups of CD-1 mice (n = 10 per group) were infected intramuscularly (IM) with rSA₁₄, rSA₁₄-14-2, or one of the six rSA₁₄-14-2 derivatives described in Fig 8, at a dose ranging from 1.5 to 1.5×10⁵ PFU/mouse. Note that the chimeric virus rSA₁₄-14-2/e, referenced in Fig 7, was included in the experiment for comparison. A group of mock-infected mice (n = 10) was also included as a control. The mice were monitored daily for clinical signs and mortality for 22 days post-infection (dpi). The collected data were used to generate dose-response survival curves, from which the 50% lethal doses (LD₅₀s) were calculated.

days. Similarly, rSA₁₄-14-2/e, included for comparison, was also weakly neuroinvasive, with an IM LD₅₀ of 2.6×10⁴ PFU and an IM RST of 9 to 13 days, as shown in Fig 7B. Overall, these results indicate that, unlike the E gene of rSA₁₄, which can convert rSA₁₄-14-2 to a neuroinvasive form, albeit weakly, the NS2A gene of rSA₁₄ lacked this ability; however, it could enhance the neuroinvasiveness of the already weakly neuroinvasive chimeric virus rSA₁₄-14-2/e, converting it to a strongly neuroinvasive form (Fig 9, compare rSA₁₄-14-2 vs. rSA₁₄-14-2/e vs. rSA₁₄-14-2/ns2a vs. rSA₁₄-14-2/e+ns2a). Consistent with this finding, the putative role of rSA₁₄ NS2A in promoting the neuroinvasiveness of rSA₁₄-14-2 was further strengthened when the prM and E genes were co-replaced with those of rSA₁₄ (Fig 9, compare rSA₁₄-14-2/prm+e vs. rSA₁₄-14-2/prm+e+ns2a). Our data also suggest that unlike the E gene, the other two structural protein genes, C and prM, of rSA₁₄ have little role in initiating or promoting viral neuroinvasion (Fig 9, compare rSA₁₄-14-2/e vs. rSA₁₄-14-2/prm+e, rSA₁₄-14-2/e+ns2a vs. rSA₁₄-14-2/prm+e+ns2a, rSA₁₄-14-2/c+e+ns2a vs. rSA₁₄-14-2/c−e+ns2a, rSA₁₄-14-2/e+ns2a vs. rSA₁₄-14-2/c+e+ns2a, and rSA₁₄-14-2/prm+e+ns2a vs. rSA₁₄-14-2/c−e+ns2a).

## Demonstration of productive peripheral infections by rSA₁₄, rSA₁₄-14-2, and their chimeric viruses with varying levels of neuroinvasiveness

A large body of research has established the following infection model [14]: When transmitted by an infected mosquito, JEV first infects mononuclear phagocytes at the bite site of the host's skin. These infected cells, possibly along with cell-free virions, then move to peripheral lymphoid organs, such as the spleen, where JEV replicates in mononuclear leukocytes. This replication causes viremia, leading to a systemic infection that spreads throughout the body, including the CNS. We therefore examined whether a selected set of rSA₁₄, rSA₁₄-14-2, and their chimeric viruses with varying degrees of neuroinvasiveness can cause productive peripheral infections prior to neuroinvasion. To this end, groups of CD-1 mice were infected by IM inoculation with each of rSA₁₄ and its derivative rSA₁₄/e, as well as rSA₁₄-14-2 and its six

derivatives (rSA$_{14}$-14-2/e, rSA$_{14}$-14-2/ns1, rSA$_{14}$-14-2/ns2a, rSA$_{14}$-14-2/e+ns1, rSA$_{14}$-14-2/e +ns2a, and rSA$_{14}$-14-2/e−ns2a), with rSA$_{14}$ and rSA$_{14}$-14-2 serving as references. For each virus, we inoculated one-tenth of its IM LD$_{50}$ if it was between 1.5 and 1.5×10$^5$ PFU (specifically 2.4×10$^3$ PFU for rSA$_{14}$-14-2/e, 4.7×10$^2$ PFU for rSA$_{14}$-14-2/e+ns1, 28 PFU for rSA$_{14}$-14-2/e+ns2a, and 15 PFU for rSA$_{14}$-14-2/e−ns2a), a maximum dose of 1.5×10$^5$ PFU if the IM LD$_{50}$ was greater than 1.5×10$^5$ PFU (specifically for rSA$_{14}$/e, rSA$_{14}$-14-2, rSA$_{14}$-14-2/ns1, and rSA$_{14}$-14-2/ns2a), or a minimum dose of 1.5 PFU if the IM LD$_{50}$ was less than 1.5 PFU (specifically for rSA$_{14}$). Following the infection, we harvested spleens and brains from four infected mice every two days for 16 days and measured viral loads as PFU per organ (Fig 10). In mice infected with the highly neuroinvasive reference rSA$_{14}$, we found low levels of viral load (ranging from 20 to 3.1×10$^3$ PFU/organ) in the spleen from day 2 until day 6, which became undetectable after day 8. In the brain, we found high levels of viral load (ranging from 50 to 1.4×10$^7$ PFU/organ) from day 6 to day 10, at which point the analysis ended. In mice infected with the non-neuroinvasive rSA$_{14}$-derived rSA$_{14}$/e, like its parent rSA$_{14}$, we also detected 30 to 2.8×10$^3$ PFU/organ in the spleen from day 2 to day 6, which fell below the detection limit after day 8; however, no virus was detected in the brain during the entire experiment. Importantly, the non-neuroinvasive reference rSA$_{14}$-14-2 and all its derivatives behaved similarly to rSA$_{14}$/e, replicating in the spleen yet producing 10 to 8.0×10$^2$ PFU/organ from day 2 to day 6, which

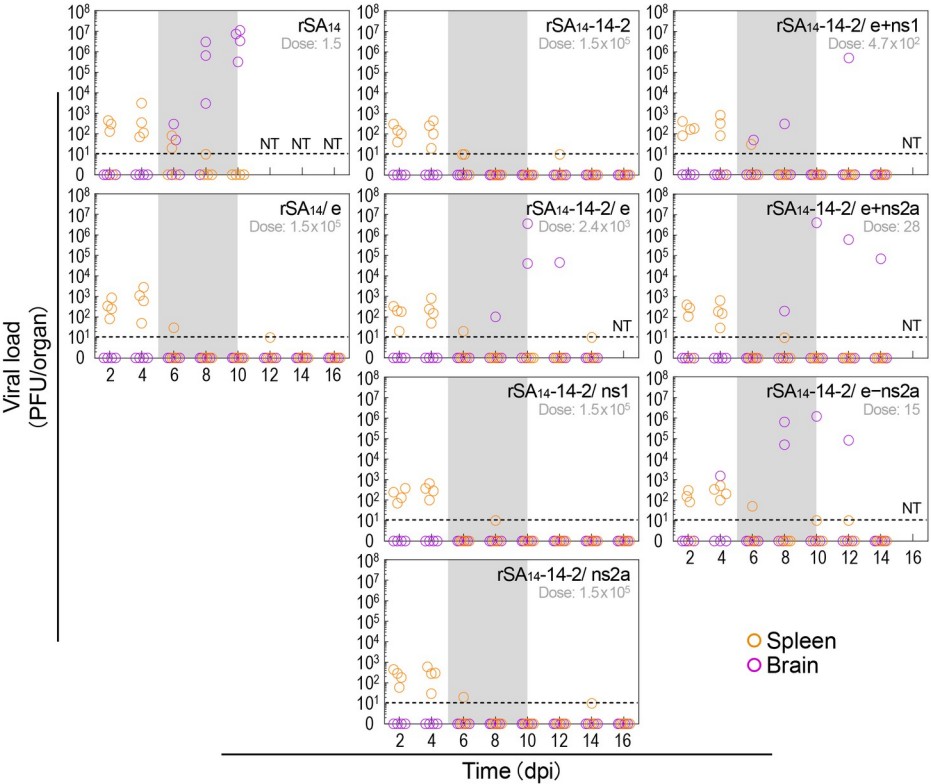

**Fig 10. Examination of peripheral infections using rSA$_{14}$, rSA$_{14}$-14-2, and their chimeric viruses of varying neuroinvasiveness.** Groups of CD-1 mice (n = 20 to 32 per group) were intramuscularly infected with rSA$_{14}$, rSA$_{14}$-14-2, or one of their derivatives. Each virus was administered at a dose of one-tenth of its IM LD$_{50}$ if between 1.5 and 1.5×10$^5$ PFU, a maximum dose of 1.5×10$^5$ PFU if greater than 1.5×10$^5$ PFU, or a minimum dose of 1.5 PFU if less than 1.5 PFU. The specific dose used for each virus is indicated. Following infection, spleens and brains were harvested from four infected mice every 2 days for 16 days, and viral loads were measured by immunoplaque assays on BHK-21 cells. Results are presented as PFU per organ. The dotted line indicates the detection limit. NT, not tested.

decreased to undetectable levels after day 8. In the brain, however, we did not detect viral replication if their IM LD$_{50}$s were >1.5×10$^5$ PFU (for rSA$_{14}$-14-2, rSA$_{14}$-14-2/ns1, and rSA$_{14}$-14-2/ns2a) or we detected it only occasionally, yielding a viral load of 50 to 4.0×10$^6$ PFU/organ, if their IM LD$_{50}$s were <1.5×10$^5$ PFU (for rSA$_{14}$-14-2/e, rSA$_{14}$-14-2/e+ns1, rSA$_{14}$-14-2/e+ns2a, and rSA$_{14}$-14-2/e–ns2a) from day 4 until the end of the experiment. Taken together, our findings indicate that rSA$_{14}$, rSA$_{14}$-14-2, and their chimeric viruses with varying levels of neuroinvasiveness can clearly establish a productive infection in the spleen at the early stage of viral infection.

## Mapping the specific amino acid residues of E, NS1/1', and NS2A that differ between SA$_{14}$ and SA$_{14}$-14-2 onto the known or predicted protein structures

The current study has provided genetic evidence demonstrating the roles of E, NS1/1', and NS2A genes in the differences in viral pathogenicity between SA$_{14}$ and SA$_{14}$-14-2. For each gene product, we mapped the position of strain-specific amino acid residues onto the known or predicted protein structure to gain insight into the structural basis for its role in viral neuroinvasiveness and/or neurovirulence (Fig 11). First, the E protein, the sole determinant of neurovirulence and the key initiator of neuroinvasion, is the viral surface glycoprotein that mediates viral entry into host cells [37,122]. On a mature JEV virion, 180 copies of the E protein form 90 antiparallel dimers, with each monomer being divided into three topological regions: the ectodomain region containing three structurally distinct domains (DI-DIII), the stem region containing three α-helices (H1-H3) on the viral membrane, and the anchor region containing two α-helices across the viral membrane [35]. We found 11 point mutations in the E gene of SA$_{14}$-14-2 compared to that of SA$_{14}$, nine of which are missense mutations located in the following domains/regions (Fig 11A): (*i*) In DI, a centrally located β-barrel structure, three missense mutations are positioned near the top of the barrel, specifically E$^{138}$K on strand E$_0$ and I$^{176}$V and T$^{177}$A on strand G$_0$ with their side chains pointing outward from the barrel. (*ii*) In DII, an elongated dimerization domain with the c-d loop (also known as the fusion loop) at one end and DI connected at the other end, four missense mutations are positioned along one side of the domain at the interface between the two E ectodomains. These include L$^{107}$F in the fusion loop with the side chain pointing downward toward the viral membrane, E$^{244}$G in the i-j loop with the side chain pointing diagonally downward toward the adjacent DI, Q$^{264}$H in helix αB at the DII-DII interface with the side chain oriented horizontally toward the neighboring DII, and K$^{279}$M on strand I at the DI-DII junction with the side chain pointing outward from the domain. (*iii*) In DIII, an immunoglobulin-like globular domain connected at one end to DI and at the other end to the stem region, one missense mutation, A$^{315}$V, is positioned on strand A in close proximity to L$^{107}$F situated in the fusion loop of the neighboring DII with the side chain pointing toward the L$^{107}$F mutation. (*iv*) In the stem region, one missense mutation, K$^{439}$R, is positioned in helix H3 underneath the ectodomain with the side chain facing upward. Notably, our map reveals that the previously proposed receptor binding pocket [35] is surrounded by three mutations: DI-E$^{138}$K and DII-K$^{279}$M from one E monomer, and DII-E$^{244}$G from the other E monomer, with their side chains all oriented toward the center of the pocket, facing either upward or downward.

NS1, identified as the first key promoter of neuroinvasiveness, plays multiple roles in viral replication, disease pathogenesis, and immune evasion by interacting with other viral proteins and cellular proteins [123–125]. This non-transmembrane glycoprotein consists of three structurally distinct domains: a central β-ladder, a dimerizing β-roll, and a protruding wing domain, with the wing domain linked to the other two domains by a small connector

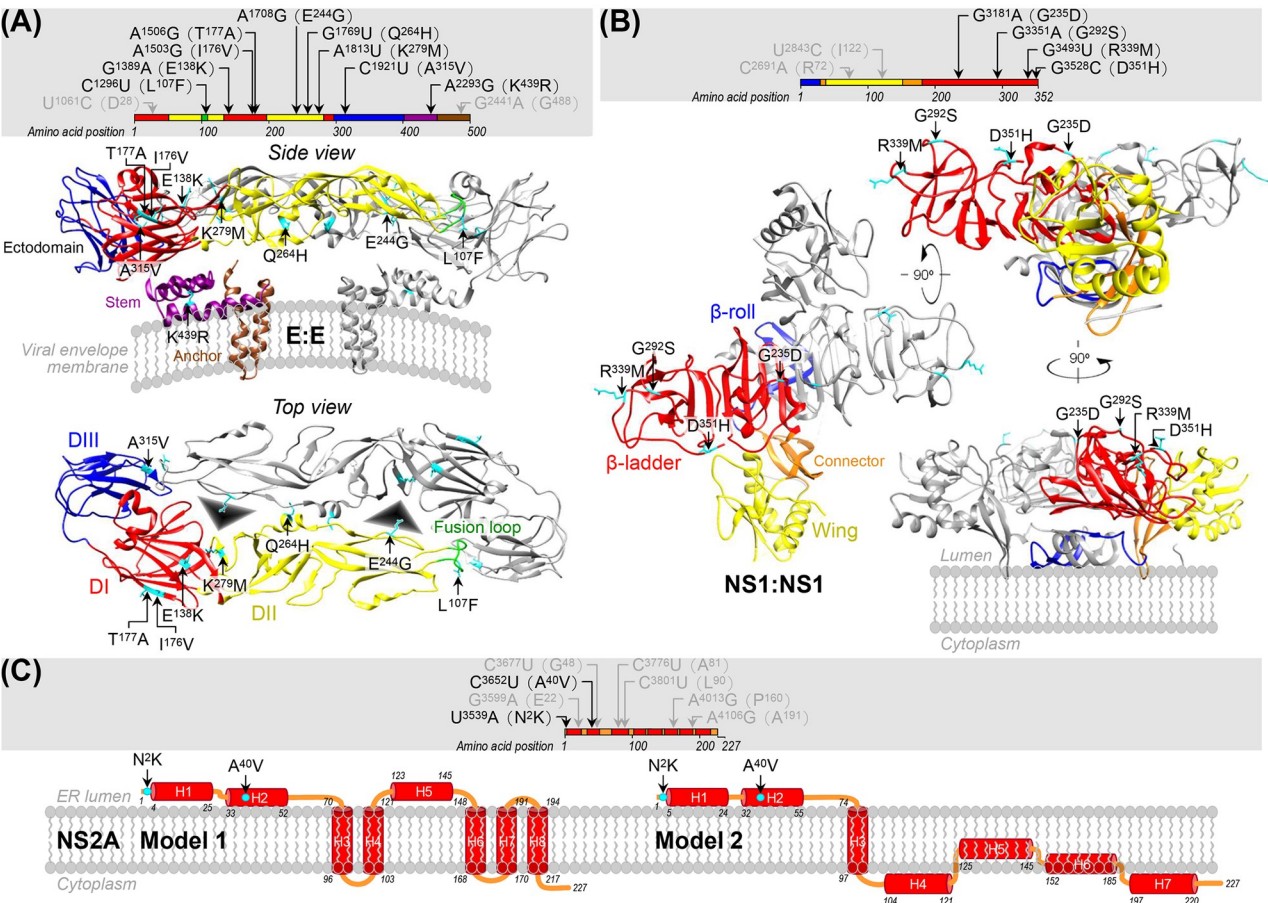

**Fig 11. Mapping strain-specific amino acid residues of proteins E, NS1, and NS2A onto their known or predicted protein structures. (A)** E. The top panel displays the locations of nine missense mutations (in black) and two silent mutations (in gray) identified within the primary sequence of the E protein of SA₁₄-14-2 (compared to that of SA₁₄). The bottom panel maps the positions of these nine missense mutations (cyan) onto the atomic structure of the E:E dimer (PDB accession code: 5WSN) located on the viral envelope membrane of the wild-type JEV strain P3 [35]. Note that all nine amino acids are identical between the P3 and SA₁₄ strains. One of the two E monomers is color-coded to represent three topological regions: the ectodomain region, which contains three structurally distinct domains (DI, red; DII, yellow; and DIII, blue); the stem region, which contains three α-helices (purple); and the anchor region, which contains two α-helices (brown). The fusion loop, located at the tip of DII, is in green. Two potential receptor binding sites [35] are depicted as black triangles. (**B**) NS1. The top panel presents the locations of four missense mutations (black) and two silent mutations (gray) found within the primary sequence of the NS1 protein of SA₁₄-14-2. The bottom panel maps the positions of these four missense mutations (cyan) onto the crystal structure of the membrane-associated NS1:NS1 dimer (PDB accession code: 4O6D) of the wild-type WNV strain NY99 [126]. Note that of the four amino acids, three (G²³⁵, G²⁹², and R³³⁹) are identical between the WNV NY99 and JEV SA₁₄ strains, while the remaining residue is N³⁵¹ in WNV NY99 and D³⁵¹ in JEV SA₁₄. One of the two NS1 monomers is color-coded to denote three structurally distinct domains: a β-ladder (red), a β-roll (blue), and a wing (yellow) domain, with a connector (orange) linking the wing domain to the other two domains. (**C**) NS2A. The top panel illustrates the locations of two missense mutations (black) and six silent mutations (gray) found in the primary sequence of the NS2A protein of SA₁₄-14-2. The bottom panel projects the positions of these two missense mutations (cyan) onto each of the two predicted structure models. These models are based on the amino acid sequence homology of the NS2A protein between JEV SA₁₄ (GenBank accession number: KU323483) and either DENV NGC (Model 1, with the GenBank accession number: KM204118 [134]) or ZIKV FSS13025 (Model 2, with the GenBank accession number: MH158236 [135]). Displayed in red barrels are the eight helices (H1-H8) predicted in Model 1 or seven helices (H1-H7) predicted in Model 2. For all three proteins mentioned above, the nucleotide positions are based on the sequence of the SA₁₄ genomic RNA, and the amino acid positions are based on the sequence of the corresponding protein.

[126,127]. The β-ladder and wing domains each have two sides with different structural compositions: one side predominantly with loops and the other with strands or a mix of strands and helices. NS1 proteins form a cross-shaped dimer within the secretory pathway compartments, by attaching the two intertwined β-roll domains and two flanking connectors to the inner leaflet of the lipid bilayer membrane and arranging the two β-ladder and two wing

domains in a cross shape over the β-roll domains with their loop-dominant sides facing the inner space [126,128,129]. These NS1 dimers further organize into a sandwich-shaped tetramer or a barrel-shaped hexamer, both of which are released from infected cells. In both structures, the β-roll domains and connectors of each dimer are oriented inward, while the loop-dominant sides of the β-ladder and wing domains face outward [130,131]. Our sequence comparison of the NS1 genes from SA$_{14}$ and SA$_{14}$-14-2 revealed that SA$_{14}$-14-2 has six point mutations, four of which are missense mutations. Remarkably, each of the four amino acid substitutions is located within a loop on the loop-dominant side of the β-ladder domain of JEV NS1, which was the only domain with a crystal structure available at the time of analysis [132]: specifically, G$^{235}$D in the unusually long β13-β14 loop (also known as the "spaghetti loop"), G$^{292}$S in the β15-β16 loop, R$^{339}$M in the β20-β21 loop, and D$^{351}$H in the unstructured region at the C-terminus of the protein, with all their side chains exposed to the exterior (S3 Fig). These positions were confirmed in the available crystal structure of the full-length WNV NS1 (Fig 11B), where the β-ladder domain closely aligns with that of JEV NS1 (S3 Fig). Therefore, our findings suggest that the four amino acid residues that differ in NS1 between SA$_{14}$ and SA$_{14}$-14-2 are exposed on the cross-shaped surface facing the interior of the secretory pathway compartments on membrane-associated dimers, and on the secreted tetramers and hexamers, they are exposed to the exterior of those multimers, facing toward the extracellular environment. In addition, NS1', which is expressed from the genome of SA$_{14}$ but not of SA$_{14}$-14-2, includes the entire 352 amino acids of NS1, the first nine amino acids of NS2A preceding the frameshift, and then the unique 43 amino acids resulting from the frameshift.

NS2A, recognized as the second key promoter of neuroinvasiveness, facilitates viral replication and counteracts the host's immune response [53,133]. NS2A is a multi-spanning transmembrane protein, yet its atomic structure remains undetermined. Membrane topology models, primarily based on biochemical and mutagenetic studies, suggest that DENV NS2A consists of eight α-helices (H1-H8). It is proposed that H1, H2, and H5 reside on the lumenal side of the ER membrane, whereas H3, H4, H6, H7, and H8 traverse the ER membrane [134]. On the other hand, ZIKV NS2A is believed to contain seven α-helices, with the first three helices sharing the same membrane topology as those of DENV NS2A. The remaining helices (fourth to seventh) are predicted to exhibit a distinct membrane topology, characterized by a specific peripheral association on the cytoplasmic side of the ER membrane [135]. A sequence comparison with SA$_{14}$ revealed eight point mutations in the NS2A gene of SA$_{14}$-14-2. Of these mutations, two alter amino acid residues within the N-terminal region of the NS2A protein, which are part of the first two α-helices proposed to be on the lumenal side of the ER membrane in both DENV and ZIKV NS2A. Specifically, the N$^{2}$K mutation is located near the N-terminus preceding H1, and the A$^{40}$V mutation is located at the beginning of H2 (Fig 11C). These findings suggest that the two amino acid changes in NS2A between SA$_{14}$ and SA$_{14}$-14-2 are likely situated on the lumenal side of the ER membrane.

## Discussion

Among the diseases caused by mosquito-borne neurotropic orthoflaviviruses, JEV infection is the only one preventable in humans through vaccination, commonly with the live-attenuated JE vaccine SA$_{14}$-14-2 [27]. This vaccine is a non-neuroinvasive and non-neurovirulent variant of the highly neuroinvasive and highly neurovirulent wild-type JEV strain SA$_{14}$ [110,111,116,136,137]. The isogenic yet pathogenetically distinct SA$_{14}$ – SA$_{14}$-14-2 virus pair provides a unique opportunity to investigate the viral factors critical for JEV pathogenicity. In the present study, we methodically analyzed the viral factors influencing the virulence and attenuation of JEV and, for the first time, utilized the unique SA$_{14}$ – SA$_{14}$-14-2 virus pair to

achieve an unprecedented depth and breadth of characterization, both *in vitro* and *in vivo*. We conducted a series of molecular, genetic, and structural mapping analyses, which included: (*i*) constructing two full-length infectious cDNA clones of SA$_{14}$ and SA$_{14}$-14-2 as BACs; (*ii*) generating 20 gene-swapped mutants between SA$_{14}$ and SA$_{14}$-14-2 in a systematic and stepwise manner; (*iii*) examining the *in vitro* virological characteristics of these mutants in hamster kidney cells, a type of cell used for the *in vitro* passage of SA$_{14}$ during its attenuation process; (*iv*) evaluating the *in vivo* pathogenic properties of the mutants in outbred mice, an animal model employed for the *in vivo* passage of SA$_{14}$ during its attenuation process; and (*v*) mapping the amino acid residues differing between SA$_{14}$ and SA$_{14}$-14-2 in viral virulence factors onto their known or predicted protein structures. Our reciprocal approach has collectively demonstrated that a single viral factor, specifically the E protein, is both necessary and sufficient to convert the highly virulent SA$_{14}$ strain into an avirulent form. In contrast, multiple viral factors, including the E, NS1/1', and NS2A proteins, are required to render the avirulent SA$_{14}$-14-2 strain highly virulent. From a mechanistic perspective, our comprehensive analysis revealed that the E protein plays a dual role in JEV pathogenicity: It acts as the master regulator of viral neurovirulence and also as the primary initiator of viral neuroinvasion. In addition, our findings indicated that the NS1/1' and NS2A proteins serve as secondary promoters, enhancing viral neuroinvasiveness following the initial neuroinvasion triggered by the E protein. In line with the roles of the E, NS1/1', and NS2A proteins in controlling JEV pathogenicity in mice, our results also highlighted their functional significance in the spread of JEV in cell culture.

During the process of infection by orthoflaviviruses, the E protein, a transmembrane glycoprotein on the viral envelope [32,35], interacts with cellular receptors on the cell surface [38–40]. This interaction is believed to initiate receptor-mediated endocytosis and low pH-dependent membrane fusion within an endosomal compartment as JEV enters the host cell, leading to the release of viral genomic RNA into the cytoplasm [37]. Although several host factors such as glycosaminoglycans and C-type lectins have been implicated in JEV entry [37], the precise host factor that directs viral internalization and membrane fusion remains unidentified. Our current study has demonstrated that the E protein plays two key roles in JEV pathogenicity: It first acts as the primary initiator for neuroinvasion and subsequently as the sole determinant for neurovirulence. These findings indicate that the viral entry step not only determines the ability of JEV to invade the CNS from a peripheral site but also its ability to cause pathological damage within the CNS. Various cell types are susceptible to JEV infection *in vitro*; however, mononuclear phagocytes outside the CNS [62–73] and neurons within the CNS [84–89] are considered the principal target cells *in vivo*. Therefore, the entry of JEV into these specific cell types is presumably critical for determining its pathogenicity, particularly concerning its neuroinvasiveness [14,80,81] and neurovirulence [14,82,83,90]. Accumulating evidence suggests that JEV may utilize both clathrin-dependent and clathrin-independent endocytic pathways, possibly in a cell type-specific manner (e.g., neuronal vs. non-neuronal cells) [138–145]. Identifying the distinct endocytic pathways employed by each of SA$_{14}$ and SA$_{14}$-14-2 strain in mononuclear phagocytes and neurons, as well as examining differences in their host cell responses following infection, would be valuable [120]. Notably, a previous study on YFV has reported that the wild-type strain Asibi primarily uses the clathrin-dependent pathway for host cell entry, while its attenuated vaccine strain 17D employs a clathrin-independent pathway [146]. This earlier study also found that the 17D strain enters host cells more efficiently than does the Asibi strain, resulting in more effective delivery of viral genomic RNA into the cytoplasm and a stronger activation of the cytokine-mediated antiviral response [146].

In the present study, we elucidated the pivotal role of the JEV E protein in initiating viral neuroinvasion and regulating neurovirulence. We identified nine amino acid residues within the E protein that differ between the SA$_{14}$ and SA$_{14}$-14-2 strains and are likely crucial for its

function. Of these nine residues, eight are grouped into three clusters, each within one of the three structurally distinct domains (DI, DII, and DIII) of the E ectodomain on the surface of an infectious JEV virion: (*i*) The DI Cluster includes three residues (E$^{138}$K, I$^{176}$V, and T$^{177}$A) near the apex of the barrel-shaped DI. (*ii*) The DII Cluster comprises four residues (L$^{107}$F, E$^{244}$G, Q$^{264}$H, and K$^{279}$M) at the interface between the two elongated DIIs, with L$^{107}$F placed in the fusion loop at one end of each DII. (*iii*) The DIII Cluster contains one residue (A$^{315}$V) in the immunoglobulin-like DIII, close to L$^{107}$F located in the fusion loop of the neighboring DII. Numerous studies have evaluated how one or more of these eight amino acid substitutions, as well as others, affect the attenuation of various wild-type JEV strains in mice [147–156]. Despite variations in experimental conditions (e.g., mouse strain, age, JEV strain, inoculum dose, and inoculation route), two consistent findings have emerged: (*a*) No single amino acid change renders the parental wild-type JEV strains as avirulent as the SA$_{14}$-14-2 vaccine strain. (*b*) The virulence of wild-type JEVs decreases with an increasing number of amino acid substitutions. Moreover, reversing any of these eight amino acid changes in the SA$_{14}$-14-2 vaccine strain to their wild-type forms can partially restore viral virulence in mice [35,119,121,157,158]. This finding suggests that multiple amino acid residues in the E ectodomain collectively influence viral pathogenicity, especially neuroinvasion and neurovirulence, likely as a result of their combined effect on the E protein's function during host cell entry in the viral replication cycle. Our current study mapped three mutations, specifically DI-E$^{138}$K and DII-K$^{279}$M from one E monomer and DII-E$^{244}$G from the other E monomer of the E:E dimer on the viral envelope, near the previously suggested receptor binding pocket [35]. These mutations, E$^{138}$K, K$^{279}$M, and E$^{244}$G, may compromise the role of the E protein in viral entry, reducing viral neuroinvasion and neurovirulence. On the other hand, the ninth residue, K$^{439}$R, which is a conservative mutation in the stem region beneath the E protein's ectodomain, is likely to have a minimal effect on viral neuroinvasion and neurovirulence. Based on our current understanding of the virus [32], its virion structure [35], and viral entry process [37], there are three possibilities for how one or more of the nine missense mutations found in the E protein of SA$_{14}$-14-2 affect its role in viral entry and pathogenicity: (*1*) They may simply hamper viral entry into a physiologically relevant target cell (e.g., mononuclear phagocytes and neurons). (*2*) They may affect the choice of viral entry pathway, which differentially alters host cell responses to viral infection and subsequently viral pathogenicity. (*3*) They may not affect the choice of viral entry pathway but may alter host cell responses to viral infection, independent of the route of viral entry, subsequently affecting viral pathogenicity. Further research is needed to determine the precise amino acid combinations in the E protein and their underlying mechanisms that influence JEV pathogenicity.

In this study, we identified the NS1/1' protein, along with the NS2A protein, as one of the key virulence factors promoting JEV neuroinvasiveness. In orthoflaviviruses, NS1 is a non-transmembrane glycoprotein [32], primarily composed of three structurally distinct domains: a β-ladder, a β-roll, and a wing domain [126,127]. After synthesis, NS1 proteins form cross-shaped dimers that associate with the inner leaflet of the lipid bilayer membrane within the secretory pathway [126,128,129]. Notably, each dimer features two β-ladder domains in a head-to-head configuration, forming a long continuous β-sheet that spans the dimer's length. This central β-sheet, comprising 18 β-strands resembling ladder rungs, has loops of various sizes protruding between the rungs, creating a unique loop-dominant surface atop the sheet. The dimers are then transported to the cell surface [159,160] or secreted as sandwich-shaped tetramers (dimers of dimers) or barrel-shaped hexamers (trimers of dimers) [161,162]. In both tetrameric and hexameric forms, the loop-dominant surface of the central β-sheet is outward-facing [130,131,161]. Our findings indicate that NS1 has four amino acid residues (G$^{235}$D, G$^{292}$S, R$^{339}$M, and D$^{351}$H) that differ between SA$_{14}$ and SA$_{14}$-14-2 strains and are likely critical

for its function. These variations are exclusively found on the loop-dominant surface of the central β-sheet of the NS1 dimer and on the external surface of the NS1 tetramer and hexamer. We demonstrated that substituting the SA$_{14}$-14-2 NS1 gene with that of SA$_{14}$, with or without NS1' expression, increases the low neuroinvasiveness of the mutant SA$_{14}$-14-2/e by approximately 2.5- to 5-fold. However, this substitution does not affect the non-neuroinvasiveness nature of the parent SA$_{14}$-14-2. These results suggest that a specific amino acid residue(s) on the loop-dominant surface of the central β-sheet of the JEV NS1/1' multimer is important for promoting viral neuroinvasiveness, rather than for initiating viral neuroinvasion.

Previous studies have demonstrated that the orthoflavivirus NS1 protein is multifunctional, playing several roles in viral replication, including promoting viral RNA replication and morphogenesis as well as contributing to viral pathogenesis and immune evasion [123–125]. In this context, the NS1 protein of various orthoflaviviruses, such as WNV, ZIKV, DENV, and YFV, has been shown to interact with the viral structural protein prM/E and two nonstructural proteins, NS4A and NS4B [163–166]. It has also been reported to interact with numerous cellular proteins, including the receptor for activated protein C kinase 1, plasma glycoprotein factor H, ribosomal protein RPL18, complement regulator vitronectin, and complement components C1q and C4 [167–174]. It is conceivable that the loop-dominant surface of the central β-sheet in the intracellular NS1 dimer interacts with viral or cellular proteins to promote JEV neuroinvasiveness. Similarly, the loop-dominant surface of the central β-sheet in the extracellular NS1 tetramer/hexamer may interact with cellular proteins to augment JEV neuroinvasiveness. Identifying the cellular proteins that bind to the loop-dominant surface of the central β-sheet of NS1 multimers and assessing their role in JEV pathogenicity is a promising direction for future research. Furthermore, the C-terminal extended frameshift product of the NS1 protein, NS1', which is expressed by JEV, WNV, and other members of the JEV serogroup, shares all 352 amino acids with the NS1 protein [41–44]. Although this study did not directly examine the contribution of NS1' to viral pathogenicity, we observed that the neuroinvasiveness level of the mutant SA$_{14}$-14-2/e was approximately two-fold higher when SA$_{14}$ NS1 was co-expressed with NS1' than when it was expressed alone. This finding is consistent with previous research indicating that NS1' plays a role in enhancing the neuroinvasiveness of JEV and WNV [43,44].

In our study, we found that the NS2A protein, along with the NS1/1' protein, acts as a secondary promoter, amplifying JEV neuroinvasiveness after E-mediated neuroinvasion. In orthoflaviviruses, NS2A is a multi-pass transmembrane protein whose atomic structure remains elusive [32]. It is believed to consist of seven to eight α-helices: The first and second helices are situated within the ER lumen, the third helix spans the ER membrane, and the remaining four to five helices exhibit a membrane topology that varies according to the models derived from studies of DENV NS2A [134] and ZIKV NS2A [135]. Upon comparing the SA$_{14}$ and SA$_{14}$-14-2 strains, we found two amino acid variations, N$^2$K and A$^{40}$V, in the N-terminal region of NS2A, which encompasses the first two α-helices. These two helices are proposed to have an identical membrane topology in both DENV NS2A and ZIKV NS2A models [134,135]. Our findings demonstrate that replacing the SA$_{14}$-14-2 NS2A gene with that of SA$_{14}$ significantly enhances the weak neuroinvasiveness of the mutant SA$_{14}$-14-2/e by approximately 100-fold; however, this replacement does not affect the non-neuroinvasiveness of the parental SA$_{14}$-14-2. These findings therefore suggest that one or both mutations, N$^2$K and A$^{40}$V, in the N-terminal region of NS2A, which is predicted to be exposed in the ER lumen, are crucial for its role in JEV neuroinvasiveness. Previous studies with several orthoflaviviruses have suggested that NS2A functions not only as a component of the viral replication complex to promote viral RNA synthesis but also as a part of the viral assembly complex, which holds together the viral genomic RNA, three structural proteins, and the viral protease to coordinate

the assembly process [134,135,175–182]. In addition, it has been suggested that an oligomeric form of DENV NS2A possesses membrane destabilizing activity and functions as a viroporin [183]. Furthermore, the NS2A protein of JEV, WNV, and DENV is suggested to play a role in antagonizing the host's innate immune responses [133]. Interestingly, recent studies on ZIKV have reported that the NS2A protein is associated with viral virulence [184] and the disruption of cortical neurogenesis in mice [185]. Although the precise role of NS2A in viral replication and pathogenesis requires further investigation, our data suggest that the N-terminal region of NS2A may interact with viral structural or nonstructural proteins, or with unidentified cellular proteins in the ER or secretory pathways that are important for promoting JEV neuroinvasiveness. In addition to the two amino acid substitutions, we noted six silent point mutations (G$^{3599}$A, C$^{3677}$U, C$^{3776}$U, C$^{3801}$U, A$^{4013}$G, and A$^{4106}$G) in the NS2A gene that differ between the SA$_{14}$ and SA$_{14}$-14-2 strains. With the exception of the mutation G$^{3599}$A located in the -1 PRF site, which is crucial for the synthesis of NS1' [42–44], the other five mutations are unlikely to affect the function of the NS2A gene, although experimental verification is needed. The same applies to the E and NS1 genes, each harboring two silent point mutations (U$^{1061}$C and G$^{2441}$A for E and C$^{2691}$A and U$^{2843}$C for NS1), whose impact on function also remains to be determined.

In summary, we employed a reverse genetics system for SA$_{14}$ and SA$_{14}$-14-2, isogenic yet pathogenetically distinct strains of JEV, to perform the first comprehensive, unbiased, reciprocal, and systematic genetic analysis of these strains. This analysis, in conjunction with structural mapping and the application of cell culture and mouse infection models relevant to the attenuation of SA$_{14}$ into SA$_{14}$-14-2, identified three key virulence factors: E, NS1/1', and NS2A. These factors are responsible for the marked differences in viral pathogenicity, especially regarding neuroinvasion, neuroinvasiveness, and neurovirulence, observed between SA$_{14}$ and SA$_{14}$-14-2. In addition, there are 29 point mutations, including eight missense mutations (two in NS2B, two in NS3, one in NS4B, and three in NS5), in the NS2B-NS5 coding region between SA$_{14}$ and SA$_{14}$-14-2. One or more of these mutations have been shown to be involved in viral neuroinvasiveness to some extent in this study, and further investigation is needed to understand their potential roles in viral neuroinvasiveness. Overall, the insights gained from this study establish a solid foundation for an in-depth understanding of the complex virulence/attenuation mechanisms of JEV and potentially other taxonomically related encephalitic orthoflaviviruses, leading to the development of an effective antiviral strategy against JEV.

## Materials and methods

### Ethics statement

All mouse infection experiments were performed in an Animal Biosafety Level 3+ facility at the USTAR Bioinnovations Campus. These experiments strictly adhered to the guidelines for the care and use of laboratory animals, as provided by the National Institutes of Health. The protocol for the animal experiments was approved by the Institutional Animal Care and Use Committee of Utah State University (Approval number: 10106). The mice were handled minimally and were euthanized when they became moribund to minimize discomfort, distress, pain, and injury.

### Cells and viruses

BHK-21 cells were cultured in alpha minimal essential medium (α-MEM) supplemented with 10% fetal bovine serum (FBS), 2 mM L-glutamine, 1× MEM vitamin solution, and 100 U/ml penicillin-streptomycin at 37˚C in a 5% CO$_2$ atmosphere, as previously described [186]. This

study utilized two strains of JEV: the SA$_{14}$-14-2 vaccine strain, retrieved from a batch of commercial vaccine vials [121], and its parental wild-type SA$_{14}$ strain, obtained from the Biodefense and Emerging Infections Resources Repository. For both cell and mouse infection experiments, original virus stocks of both SA$_{14}$ and SA$_{14}$-14-2 were prepared by amplifying them twice in BHK-21 cells. Recombinant viruses, such as rSA$_{14}$, rSA$_{14}$-14-2, and their derivatives, were recovered from their respective full-length infectious cDNA clones in BHK-21 cells. All cell culture media and reagents were purchased from Gibco (Carlsbad, CA).

## Functional cDNAs of the parental SA$_{14}$ and SA$_{14}$-14-2 viruses

The full-length infectious cDNAs of SA$_{14}$ (designated pBac/SA$_{14}$) and SA$_{14}$-14-2 (designated pBac/SA$_{14}$-14-2) were assembled into the BAC plasmid pBeloBac11 (Fig 2), utilizing the cloning strategy previously described [121]. For both constructs, the viral genomic RNA, purified from each virus using Trizol LS reagent from Invitrogen (Gaithersburg, MD), served as the template for the synthesis of four overlapping cDNA fragments (F1 to F4). This synthesis was performed using a two-step RT-PCR method. Specific primers for the RT and PCR reactions are detailed in Table 1: JS1rt and JS1fw+JS1rv for F1, JS2rt and JS2fw+JS2rv for F2, JS3rt and

**Table 1. Sequences and directions of the oligonucleotides used in this study.**

| Oligonucleotide | Sequence (5' to 3')ª | Direction |
|---|---|---|
| JS1rt | TAGGGATCTGGGCGTTTCTGGCAAAT | Reverse |
| JS1fw | aatcccgggAGAAGTTTATCTGTGTGAACTT | Forward |
| JS1rv | attgcggccgcCCACGTCGTTGTGCACGAAGAT | Reverse |
| JS2rt | TTCTGCCTACTCTGCCCCTCCGTTGA | Reverse |
| JS2fw | aatcccgggTCAAGCTCAGTGATGTTAACAT | Forward |
| JS2rv | attgcggccgcGATGGGTTTCCGAGGATGACTC | Reverse |
| JS3rt | ACGGTCTTTCCTTCTGCTGCAGGTCT | Reverse |
| JS3fw | aatcccgggGAGGATACATTGCTACCAAGGT | Forward |
| JS3rv | attgcggccgcGTAAGTCAGTTCAATTATGGCT | Reverse |
| JS4rt | AGATCCTGTGTTCTTCCTCACCACCA | Reverse |
| JS4fw | aatcccgggAGTGGAAGGCTCAGGCGTCCAA | Forward |
| JS4rv | attgcggccgcAGATCCTGTGTTCTTCCTCACC | Reverse |
| JSsp6fw | catacccgcgtattcccacta | Forward |
| JSsp6rv | ACAGATAAACTTCTctatagtgtcccctaaa | Reverse |
| JSF1fw | aggggacactatagAGAAGTTTATCTGTGTG | Forward |
| JSF1rv | TGGATCATTGCCCATGGTAAGCTTA | Reverse |
| JSX1fw | CGAATGGATCGCACAGTGTGGAGAG | Forward |
| JSX1rv | AAAGCTTCAAACTCAAGATACCGTGCTCC | Reverse |
| JSX2fw | GGAGCACGGTATCTTGAGTTTGAAGCTTT | Forward |
| JSX2rv | cacgtggacgagggcatgcctgcag | Reverse |
| JSROfw | CCAGGAGGACTGGGTTACCAAAGCC | Forward |
| JSROrv | agggcggccgctctagAGATCCTGTGTTCTTCCTCACCAC | Reverse |
| SAfw | ATCCAACTCAACCGCAAGTC | Forward |
| SArv | TCTAAGATGGTGGGTTTCACG | Reverse |
| SAprb | FAM-CATCTCTGAAATGGGGGCCA-BHQ1 | Forward |
| BHKfw | *actggcattgtgatggactc* | Forward |
| BHKrv | *catgaggtagtctgtcaggtc* | Reverse |
| BHKprb | HEX-*ccagccaggtccagacgcagg*-BHQ2 | Reverse |

ª Viral sequences are denoted by uppercase letters, and β-actin sequences are represented by lowercase letters in italics.

JS3fw+JS3rv for F3, and JS4rt and JS4fw+JS4rv for F4. Subsequently, each of the four amplicons was subcloned by ligating the *Not*I-*Pme*I fragment of the vector pBac/PRRSV/FL [187] with the *Sma*I-*Not*I fragment of the insert F1, F2, F3, and F4 amplicons, resulting in the creation of pBac/F1, pBac/F2, pBac/F3, and pBac/F4, respectively.

The pBac/F1 subclone was modified by inserting the SP6 promoter immediately upstream of the 5'-end of the viral genome using an overlapping extension PCR method: (*i*) Two DNA fragments were amplified: one derived from the PCR of pBac$^{SP6}$/JVFLx/*Xba*I [186] using the primer pair JSsp6fw+JSsp6rv (where JSsp6rv includes the reverse sequence of the SP6 promoter), and the other derived from the PCR of pBac/F1 using the primer pair JSF1fw+JSF1rv. (*ii*) These two amplicons were then fused by PCR using the primer pair JSsp6fw+JSF1rv. (*iii*) The resulting fused amplicons were subsequently used to replace the corresponding region in pBac/F1, utilizing *Pac*I and *Bsi*WI, to create pBac/F1$^{SP6}$.

The pBac/F3 subclone was engineered by introducing a silent point mutation, A$^{9134}$→T, in the NS5 protein-coding region to inactivate the pre-existing *Xba*I site at position 9131 using an overlapping extension PCR method: (*i*) Two DNA fragments were amplified by PCR from pBac/F3 using two primer pairs, JSX1fw+JSX1rv and JSX2fw+JSX2rv. (*ii*) These two amplicons were then fused by PCR using the primer pair JSX1fw+JSX2rv. (*iii*) The resulting fused amplicons were subsequently used to replace the corresponding region in pBac/F3, utilizing *Avr*II and *Not*I, to create pBac/F3$^{KO}$.

The pBac/F4 subclone was modified by inserting a new *Xba*I run-off site immediately downstream of the 3'-end of the viral genome through site-directed mutagenesis: A DNA fragment was first amplified by PCR from pBac/F4 using the primer pair JSROfw+JSROrv (where JSROrv includes the reverse sequence of *Xba*I and *Not*I sites consecutively). The resulting amplicons were then used to replace the corresponding region in pBac/F4, utilizing *Not*I and *Sfi*I, to create pBac/F4$^{RO}$.

Finally, a full-length cDNA was constructed by sequentially ligating the 7456-bp *Not*I-*Pac*I fragment of vector pBac$^{SP6}$/JVFLx/*Xba*I [186] with the 2022-bp *Pac*I-*Bsr*GI fragment of pBac/F1$^{SP6}$, the 3689-bp *Bsr*GI-*Bam*HI fragment of pBac/F2, the 3800-bp *Bam*HI-*Ava*I fragment of pBac/F3$^{KO}$, and the 1607-bp *Ava*I-*Not*I fragment of pBac/F4$^{RO}$. The complete viral genome sequences in both pBac/SA$_{14}$ and pBac/SA$_{14}$-14-2 were verified by Sanger sequencing. For our cDNA cloning experiments, we used oligonucleotides obtained from Integrated DNA Technologies (IDT, San Diego, CA), Superscript III reverse transcriptase procured from Invitrogen (Gaithersburg, MD), Advantage HD polymerase purchased from Clontech (Mountain View, CA), and restriction enzymes obtained from New England Biolabs (NEB, Ipswich, MA).

## Functional cDNAs of chimeric viruses derived from SA$_{14}$ and SA$_{14}$-14-2

Four panels of full-length infectious cDNAs were cloned to create chimeric viruses by exchanging various viral genomic regions between pBac/SA$_{14}$ and pBac/SA$_{14}$-14-2. For our cDNA cloning procedures, we used synthetic DNAs obtained from IDT and restriction enzymes purchased from NEB.

(*i*) **Panel 1**, *depicted in* Fig 4, *includes the following six constructs*: (*a*) pBac/SA$_{14}$/5'+3'ncr was constructed by ligating the 8276-bp *Aat*II-*Pme*I fragment of pBac/SA$_{14}$-14-2 with the 5512-bp *Pme*I-*Bam*HI and 4786-bp *Bam*HI-*Aat*II fragments of pBac/SA$_{14}$. (*b*) pBac/SA$_{14}$-14-2/5'+3'ncr was constructed by ligating the 8276-bp *Aat*II-*Pme*I fragment of pBac/SA$_{14}$ with the 5512-bp *Pme*I-*Bam*HI and 4786-bp *Bam*HI-*Aat*II fragments of pBac/SA$_{14}$-14-2. (*c*) pBac/SA$_{14}$/c–ns2a was constructed by ligating the 4271-bp *Pac*I-*Eco*47III fragment of pBac/SA$_{14}$-14-2/5'+3'ncr with the 4026-bp *Eco*47III-*Mlu*I and 10277-bp *Mlu*I-*Pac*I fragments of pBac/SA$_{14}$. (*d*) pBac/SA$_{14}$-14-2/c–ns2a was constructed by ligating the 4271-bp *Pac*I-*Eco*47III fragment of

pBac/SA$_{14}$/5'+3'ncr with the 4026-bp *Eco*47III-*Mlu*I and 10277-bp *Mlu*I-*Pac*I fragments of pBac/SA$_{14}$-14-2. (*e*) pBac/SA$_{14}$/e–ns2a was constructed by ligating the 1064-bp *Pac*I-*Mlu*I fragment of pBac/SA$_{14}$ with the 4647-bp *Mlu*I-*Bam*HI and 12863-bp *Bam*HI-*Pac*I fragments of pBac/SA$_{14}$/c–ns2a. (*f*) pBac/SA$_{14}$-14-2/e–ns2a was constructed by ligating the 1064-bp *Pac*I-*Mlu*I fragment of pBac/SA$_{14}$-14-2 with the 4647-bp *Mlu*I-*Bam*HI and 12863-bp *Bam*HI-*Pac*I fragments of pBac/SA$_{14}$-14-2/c–ns2a.

(*ii*) **Panel 2**, *depicted in* Fig 6, *includes the following six constructs*: (*a*) pBac/SA$_{14}$/e+ns1 +prf$^{Neg}$ was created by joining four fragments: the 412-bp *Bsp*EI-*Avr*II fragment of synthetic DNA A (including the 3'-terminal 89 bp of the SA$_{14}$-14-2 NS1 gene and the 5'-terminal 323 bp of the SA$_{14}$ NS2A gene with the G$^{3599}$A point mutation), the 1719-bp *Avr*II-*Bam*HI fragment of pBac/SA$_{14}$, the 12863-bp *Bam*HI-*Pac*I fragment of pBac/SA$_{14}$, and the 3580-bp *Pac*I-*Bsp*EI fragment of pBac/SA$_{14}$/e–ns2a. (*b*) pBac/SA$_{14}$-14-2/e+ns1+prf$^{Pos}$ was created by joining four fragments: the 412-bp *Bsp*EI-*Avr*II fragment of synthetic DNA B (including the 3'-terminal 89 bp of the SA$_{14}$ NS1 gene and the 5'-terminal 323 bp of the SA$_{14}$-14-2 NS2A gene with the A$^{3599}$G point mutation), the 1719-bp *Avr*II-*Bam*HI fragment of pBac/SA$_{14}$-14-2, the 12863-bp *Bam*HI-*Pac*I fragment of pBac/SA$_{14}$-14-2, and the 3580-bp *Pac*I-*Bsp*EI fragment of pBac/ SA$_{14}$-14-2/e–ns2a. (*c*) pBac/SA$_{14}$/e was created by joining four fragments: the 1609-bp *Mlu*I-*Apa*LI fragment of pBac/SA$_{14}$/e+ns1+prf$^{Neg}$, the 3038-bp *Apa*LI-*Bam*HI fragment of pBac/ SA$_{14}$, the 12863-bp *Bam*HI-*Pac*I fragment of pBac/SA$_{14}$, and the 1064-bp *Pac*I-*Mlu*I fragment of pBac/SA$_{14}$. (*d*) pBac/SA$_{14}$-14-2/e was created by joining four fragments: the 1609-bp *Mlu*I-*Apa*LI fragment of pBac/SA$_{14}$-14-2/e+ns1+prf$^{Pos}$, the 3038-bp *Apa*LI-*Bam*HI fragment of pBac/SA$_{14}$-14-2, the 12863-bp *Bam*HI-*Pac*I fragment of pBac/SA$_{14}$-14-2, and the 1064-bp *Pac*I-*Mlu*I fragment of pBac/SA$_{14}$-14-2. (*e*) pBac/SA$_{14}$/ns1+prf$^{Neg}$ was created by joining four fragments: the 3038-bp *Apa*LI-*Bam*HI fragment of pBac/SA$_{14}$/e+ns1+prf$^{Neg}$, the 12863-bp *Bam*HI-*Pac*I fragment of pBac/SA$_{14}$, the 1064-bp *Pac*I-*Mlu*I fragment of pBac/SA$_{14}$, and the 1609-bp *Mlu*I-*Apa*LI fragment of pBac/SA$_{14}$. (*f*) pBac/SA$_{14}$-14-2/ns1+prf$^{Pos}$ was created by joining four fragments: the 3038-bp *Apa*LI-*Bam*HI fragment of pBac/SA$_{14}$-14-2/e+ns1+prf$^{Pos}$, the 12863-bp *Bam*HI-*Pac*I fragment of pBac/SA$_{14}$-14-2, the 1064-bp *Pac*I-*Mlu*I fragment of pBac/SA$_{14}$-14-2, and the 1609-bp *Mlu*I-*Apa*LI fragment of pBac/SA$_{14}$-14-2.

(*iii*) **Panel 3**, *depicted in* Fig 8, *includes the following six constructs*: (*a*) pBac/SA$_{14}$-14-2/ns2a was generated by ligating the 3511-bp *Bsi*WI-*Eco*47III fragment of pBac/SA$_{14}$/e+ns1+prf$^{Neg}$ with the 4026-bp *Eco*47III-*Mlu*I and 11037-bp *Mlu*I-*Bsi*WI fragments of pBac/SA$_{14}$-14-2. (*b*) pBac/SA$_{14}$-14-2/e+ns2a was generated by ligating the 2867-bp *Bsi*WI-*Nsi*I fragment of pBac/ SA$_{14}$-14-2/e with the 4670-bp *Nsi*I-*Mlu*I and 11037-bp *Mlu*I-*Bsi*WI fragments of pBac/SA$_{14}$-14-2/ns2a. (*c*) pBac/SA$_{14}$-14-2/prm+e+ns2a was generated by combining the 760-bp *Pac*I-*Bsi*WI fragment of synthetic DNA C (including the SP6 promoter, the 5'NCR and C gene of SA$_{14}$-14-2, and the prM gene of SA$_{14}$) with the 4951-bp *Bsi*WI-*Bam*HI and 12863-bp *Bam*HI-*Pac*I fragments of pBac/SA$_{14}$-14-2/e+ns2a. (*d*) pBac/SA$_{14}$-14-2/c+e+ns2a was generated by combining the 760-bp *Pac*I-*Bsi*WI fragment of synthetic DNA D (including the SP6 promoter, the 5'NCR of SA$_{14}$-14-2, the C gene of SA$_{14}$, and the prM gene of SA$_{14}$-14-2) with the 4951-bp *Bsi*WI-*Bam*HI and 12863-bp *Bam*HI-*Pac*I fragments of pBac/SA$_{14}$-14-2/e+ns2a. (*e*) pBac/ SA$_{14}$-14-2/c–e+ns2a was generated by combining the 760-bp *Pac*I-*Bsi*WI fragment of synthetic DNA E (including the SP6 promoter, the 5'NCR of SA$_{14}$-14-2, and the C and prM genes of SA$_{14}$) with the 4951-bp *Bsi*WI-*Bam*HI and 12863-bp *Bam*HI-*Pac*I fragments of pBac/SA$_{14}$-14-2/e+ns2a. (*f*) pBac/SA$_{14}$-14-2/prm+e was generated by ligating the 3627-bp *Pac*I-*Nsi*I fragment of pBac/SA$_{14}$-14-2/prm+e+ns2a with the 4670-bp *Nsi*I-*Mlu*I and 10277-bp *Mlu*I-*Pac*I fragments of pBac/SA$_{14}$-14-2.

(*iv*) **Panel 4**, *depicted in* S1 Fig, *includes the following two constructs*: (*a*) pBac/SA$_{14}$-14-2/e +ns1 was constructed by combining four fragments: the 412-bp *Bsp*EI-*Avr*II fragment of

synthetic DNA F (including the 3'-terminal 89 bp of the SA$_{14}$ NS1 gene and the 5'-terminal 323 bp of the SA$_{14}$-14-2 NS2A gene), the 1719-bp *Avr*II-*Bam*HI, 12863-bp *Bam*HI-*Pac*I, and 3580-bp *Pac*I-*Bsp*EI fragments of pBac/SA$_{14}$-14-2/e+ns1+prf$^{Pos}$. (*b*) pBac/SA$_{14}$-14-2/ns1 was constructed by combining four fragments: the 412-bp *Bsp*EI-*Avr*II fragment of synthetic DNA F (including the 3'-terminal 89 bp of the SA$_{14}$ NS1 gene and the 5'-terminal 323 bp of the SA$_{14}$-14-2 NS2A gene), the 1719-bp *Avr*II-*Bam*HI, 12863-bp *Bam*HI-*Pac*I, and 3580-bp *Pac*I-*Bsp*EI fragments of pBac/SA$_{14}$-14-2/ns1+prf$^{Pos}$.

### *In vitro* run-off transcription

Viral genomic RNAs were synthesized by *in vitro* run-off transcription from a full-length JEV cDNA template. Initially, the BAC plasmid, which contains a full-length JEV cDNA, was linearized by digestion with *Xba*I (NEB) and subsequently treated with mung bean nuclease (NEB). This linearized plasmid was purified via phenol-chloroform extraction and ethanol precipitation. The purified plasmid was then used as a DNA template for *in vitro* run-off transcription, as previously described [121,186]. The transcription reaction was performed using SP6 RNA polymerase (NEB) with the RNA cap structure analog m$^7$GpppA (NEB) at 37°C for 1 h, according to the manufacturer's instructions. After transcription, the quality of the RNA transcripts was assessed using 0.6% agarose gel electrophoresis and a NanoDrop 2000 spectrophotometer (Thermo Scientific, Waltham, MA).

### RNA transfection/infectious center assay

The specific infectivity of viral RNA, synthesized *in vitro* from a full-length JEV cDNA, was assessed by an infectious center assay. This assay involved introducing the viral RNA into BHK-21 cells via electroporation, using an ECM 830 electroporator (BTX, San Diego, CA) as detailed in our earlier reports [121,186]. In brief, cells were collected through trypsin digestion, washed three times with cold phosphate-buffered saline (PBS), and then resuspended in the same cold PBS. An aliquot of $8\times10^6$ cells was electroporated in 400 μl of cold PBS with 2 μg of the viral RNA under optimized conditions: a voltage setting of 980 V, a pulse length of 99 μs, and a sequence of 5 pulses. After a 10-min incubation at room temperature to allow them to recover from the electric shocks, 10-fold serial dilutions of the electroporated cells were placed onto monolayers of unelectroporated cells ($5\times10^5$ cells/well) in a six-well plate. After incubating for 4–6 h to allow the cells to attach to the wells, the cells were overlaid with 3 ml of α-MEM containing 10% FBS and 0.5% SeaKem LE agarose (FMC BioProducts, Rockland, ME). The plates were then incubated at 37°C in a 5% CO$_2$ environment for 4 days. The cell monolayers were fixed with 7% formaldehyde. Infectious centers of plaques were visualized by immunostaining (see below).

### Recovery of recombinant viruses

A total of $8\times10^6$ BHK-21 cells were transfected with 10 μg of viral RNA, transcribed *in vitro* from a full-length JEV cDNA, as described above. The transfected cells were then seeded into a 150-mm culture dish at a density of $4\times10^6$ cells/dish and cultured for 1–2 days in complete culture medium at 37°C in a 5% CO$_2$ incubator. Once 75% of the cells exhibited cytopathic effects due to viral replication, the culture supernatant containing recombinant viruses was collected. Any cell debris present in the collected supernatant was removed by centrifugation at 2,000 × g for 10 min at 4°C. The cleared supernatant was then aliquoted and stored at -80°C for future use.

## Reverse transcription-quantitative PCR (RT-qPCR)

Total cellular RNAs were extracted from BHK-21 cells, which were either mock-infected or infected with SA$_{14}$, rSA$_{14}$, SA$_{14}$-14-2, rSA$_{14}$-14-2, or one of their derivatives using Trizol reagent (Invitrogen). A two-step RT-qPCR was performed in duplicate to quantify the viral genomic RNA and the cellular β-actin mRNA, which served as an internal control, according to our previously established protocol [116]. Initially, equal amounts of the total cellular RNAs were used for the RT reactions, using Superscript III reverse transcriptase (Invitrogen) at 50˚C for 30 min, followed by heat inactivation at 70˚C for 15 min. Subsequently, equal volumes of the RT reactions were used to perform the qPCR reactions, using the TaqMan Gene Expression Master Mix (Applied Biosystems, Foster City, CA) on a 7500 Fast Real-Time PCR System (Applied Biosystems). The thermal cycle began with an initial incubation of 10 min at 95˚C, followed by 40 cycles of 15 sec at 95˚C and 1 min at 60˚C. Relative quantification values were calculated using the $2^{-\Delta\Delta CT}$ method and expressed as fold change over time or at an indicated time point relative to the 6-h time point post-infection [116]. The primer pairs and fluorogenic probes specified in Table 1 for this assay include SAfw+SArv and SAprb for the JEV NS3 gene as well as BHKfw+BHKrv and BHKprb for the BHK-21 β-actin gene.

## Immunoblot analysis

Total cell lysates were prepared from BHK-21 cells infected with SA$_{14}$, rSA$_{14}$, SA$_{14}$-14-2, rSA$_{14}$-14-2, or one of their derivatives, as well as from mock-infected cells. The lysis buffer contained 80 mM Tris-HCl (pH 6.8), 2% sodium dodecyl sulfate (SDS), 10% glycerol, 100 mM dithiothreitol, and 0.2% bromophenol blue. Equal amounts of the total cell lysates were boiled for 5 min, separated on an SDS-polyacrylamide gel using a Mini Protean Vertical Electrophoresis System (Bio-Rad, Hercules, CA), and transferred from the gel to a polyvinylidene difluoride membrane using a Trans-Blot SD Semi-Dry Transfer Cell (Bio-Rad). The membrane was then subjected to immunoblotting as previously described [41]. The primary antibodies used included a mouse anti-JEV hyperimmune antiserum (diluted 1:1,000), a rabbit anti-JEV NS1 antiserum (diluted 1:1,000), a rabbit anti-JEV NS1' antiserum (diluted 1:1,000), and a mouse anti-glyceraldehyde 3-phosphate dehydrogenase (GAPDH) antibody (diluted 1:10,000). The secondary antibodies used were alkaline phosphatase (AP)-conjugated goat anti-mouse and anti-rabbit IgGs (diluted 1:5,000; Jackson ImmunoResearch, West Grove, PA). The antibody-reactive proteins on the membrane were detected using 5-bromo-4-chloro-3-indolyl phosphate and nitroblue tetrazolium (Sigma-Aldrich, St. Louis, MO) as substrates for AP.

## Viral growth analysis

A monolayer of BHK-21 cells, pre-seeded in a 60-mm culture dish for 18 h, was infected with SA$_{14}$, rSA$_{14}$, SA$_{14}$-14-2, or rSA$_{14}$-14-2 at an MOI of 1 for 1 h. After infection, the cell monolayer was washed once with 5 ml of α-MEM and then incubated in 5 ml of complete culture medium at 37˚C in a 5% CO$_2$ incubator. At 6, 12, 18, 24, 36, 48, and 72 h post-infection, a portion of the culture supernatant was collected for virus titration. For the titration, monolayers of BHK-21 cells, pre-seeded in a 6-well plate at a density of $3\times10^5$ cells/well for 18 h, were infected with serial 10-fold dilutions of the culture supernatant, which contained the virus, for 1 h. After infection, the cell monolayers were each overlaid with 3 ml of α-MEM supplemented with 10% FBS and 0.5% SeaKem LE agarose. Four days after infection, viral plaques were visualized by immunostaining (see below), and viral titers were determined as PFU/ml.

## Plaque immunostaining

Viral plaques formed on BHK-21 cell monolayers were immunostained according to our previously established protocol [116]. In brief, the cell monolayer was fixed with 7% formaldehyde and permeabilized with 0.25% Triton X-100. Subsequently, it was stained with a rabbit anti-JEV NS3 antiserum at a 1:1,000 dilution. After the primary antibody treatment, the cell monolayer was again incubated with a horseradish peroxidase (HRP)-conjugated goat anti-rabbit IgG (Jackson ImmunoResearch) at a 1:1,000 dilution. The plaques on the cell monolayer that reacted with the NS3 antibody were detected using 3,3'-diaminobenzidine (Vector Laboratories, Burlingame, CA) as the substrate for HRP. The average plaque size was determined in mm by analyzing 10 randomly picked plaques.

## Mouse infection

Three-week-old female CD-1 mice were purchased from Charles River Laboratories (Wilmington, MA). The mice were inoculated either intramuscularly (50 µl total volume) or intracerebrally (20 µl total volume) with 10-fold serial dilutions of SA$_{14}$, rSA$_{14}$, SA$_{14}$-14-2, rSA$_{14}$-14-2, or one of their derivatives in α-MEM. As a control, mice were mock-inoculated with the same total volume of the culture supernatant of BHK-21 cells, either intramuscularly or intracerebrally. Following infection, the mice were monitored for clinical signs or death once or twice daily over 22 days. Dose-response survival curves for each virus were generated using the Kaplan-Meier method, and LD$_{50}$ values were estimated using the Reed-Muench method, as described in our previous report [116]. To determine peripheral infections caused by rSA$_{14}$, rSA$_{14}$-14-2, and their derivatives, mice were intramuscularly infected with each virus at a dose of one-tenth of its IM LD$_{50}$ if between 1.5 and $1.5 \times 10^5$ PFU, a maximum dose of $1.5 \times 10^5$ PFU if greater than $1.5 \times 10^5$ PFU, or a minimum dose of 1.5 PFU if less than 1.5 PFU. Every 2 days after infection for 16 days, four infected mice were transcardially perfused with cold PBS, and their spleens and brains were harvested. The collected organs were homogenized in 1 ml of cold PBS using the TissueLyser II system (Qiagen, Valencia, CA) as recommended by the manufacturer, followed by centrifugation at $10,000 \times g$ for 10 min at 4˚C. Viral loads in the cleared homogenates were measured by plaque assays on BHK-21 cells, combined with immunostaining of viral plaques (see above). Viral loads were determined as PFU/organ.

## Sequence- and structure-based analyses

The complete genome sequences of JEV SA$_{14}$ (GenBank accession number: KU323483) and SA$_{14}$-14-2 (GenBank accession number: JN604986) were aligned using the BLAST blastn program, which is accessible via the NCBI web server at https://blast.ncbi.nlm.nih.gov/Blast.cgi. In addition, the amino acid sequence of the NS2A gene of JEV SA$_{14}$ (GenBank accession number: KU323483) was compared with those of DENV NGC (GenBank accession number: KM204118) and ZIKV FSS13025 (GenBank accession number: MH158236) using the BLAST blastp program, which is accessible via the NCBI web server at https://blast.ncbi.nlm.nih.gov/Blast.cgi. The RNA structures, located at nucleotide positions 3551–3630 in the JEV SA$_{14}$ and SA$_{14}$-14-2 genomes, were predicted using the IPknot program, which is accessible through the Rtips web server at http://ws.sato-lab.org/rtips/. The atomic structures of the full-length E protein dimer of JEV P3 (PDB accession code: 5WSN), the full-length NS1 protein dimer of WNV NY99 (PDB accession code: 4O6D), and the C-terminal NS1 β-ladder domain of JEV SA$_{14}$ (PDB accession code: 5O19) were visualized using the UCSF Chimera 1.10.2, which is downloadable from the website https://www.cgl.ucsf.edu/chimera/.

## Supporting information

**S1 Fig. Virological properties of three rSA$_{14}$-14-2 derivatives *in vitro*.** (**A**) Schematic diagram showing the genomes of rSA$_{14}$-14-2 and its three derivatives (rSA$_{14}$-14-2/e–ns2a, rSA$_{14}$-14-2/e+ns1, and rSA$_{14}$-14-2/ns1). Colors in the diagram represent the gene(s) of rSA$_{14}$-14-2 replaced with the corresponding gene(s) of rSA$_{14}$: rSA$_{14}$, red; rSA$_{14}$-14-2, blue. (**B-D**) BHK-21 cells were either mock-infected or infected with rSA$_{14}$-14-2 or one of its three derivatives at an MOI of 1. (**B**) Viral RNA replication. The levels of viral genomic RNA at 20 h post-infection were compared to those at 6 h post-infection by RT-qPCR of total cellular RNAs using a JEV NS3-specific TaqMan probe. The results are presented as fold changes. (**C**) Viral protein production. The levels of viral proteins at 20 h post-infection were determined by immunoblotting of total cell lysates using a mouse anti-JEV hyperimmune antiserum. GAPDH was used as a loading control. Each blot shows protein molecular weight markers in kDa on the left and viral proteins on the right. (**D**) Viral plaque morphology. Viral plaques were visualized by immunostaining cell monolayers with a rabbit anti-JEV NS3 antiserum, following a 4-day incubation period under a semi-solid overlay. Representative plaques are shown with their average dimensions (in mm).
(TIF)

**S2 Fig. Viral pathogenicity of three rSA$_{14}$-14-2 derivatives *in vivo*.** Groups of CD-1 mice were either mock-infected or infected intramuscularly (IM, n = 10 per group) or intracerebrally (IC, n = 5 per group) with rSA$_{14}$-14-2 or one of its three derivatives detailed in S1 Fig, at a dose ranging from 1.5 to $1.5 \times 10^5$ PFU/mouse. To generate dose-response survival curves, the mice were observed daily for clinical signs and mortality for 22 days after infection. The 50% lethal doses (LD$_{50}$s) were then calculated from these survival curves. dpi, days post-infection.
(TIF)

**S3 Fig. Map of four strain-specific amino acid residues of the NS1 protein onto the currently available crystal structure of its β-ladder domain.** The top panel shows the locations of four missense mutations (in black) and two silent mutations (in gray) found in the primary sequence of the NS1 protein of SA$_{14}$-14-2, as compared to that of SA$_{14}$. The nucleotide positions are based on the sequence of the SA$_{14}$ genomic RNA, and the amino acid positions are based on the sequence of the NS1 protein. The bottom panel maps the positions of the four missense mutations (cyan) onto the crystal structure of the C-terminal β-ladder domain (red, PDB accession code: 5O19) of JEV SA$_{14}$ [132]. For comparison, the crystal structure of the JEV SA$_{14}$ β-ladder domain is superimposed on the crystal structure of the full-length NS1 dimer (gray, PDB accession code: 4O6D) of WNV NY99 [126].
(TIF)

## Acknowledgments

We would like to express our gratitude to Dr. Deborah McClellan for her invaluable contribution to editing the manuscript.

## Author Contributions

**Conceptualization:** Young-Min Lee.

**Data curation:** Byung-Hak Song, Sang-Im Yun, Young-Min Lee.

**Formal analysis:** Byung-Hak Song, Sang-Im Yun, Joseph L. Goldhardt, Jiyoun Kim.

**Funding acquisition:** Young-Min Lee.

**Investigation:** Byung-Hak Song, Sang-Im Yun, Joseph L. Goldhardt, Jiyoun Kim.

**Methodology:** Byung-Hak Song, Sang-Im Yun, Joseph L. Goldhardt, Jiyoun Kim, Young-Min Lee.

**Project administration:** Young-Min Lee.

**Resources:** Sang-Im Yun, Young-Min Lee.

**Supervision:** Young-Min Lee.

**Validation:** Byung-Hak Song, Sang-Im Yun, Young-Min Lee.

**Visualization:** Byung-Hak Song, Sang-Im Yun.

**Writing – original draft:** Byung-Hak Song, Sang-Im Yun.

**Writing – review & editing:** Young-Min Lee.

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
