## [Decision Letter · Decision Letter 0]

20 Aug 2024

Dear Dr. Lee,

Thank you very much for submitting your manuscript "Key virulence factors responsible for differences in pathogenicity between clinically proven live-attenuated Japanese encephalitis vaccine SA14-14-2 and its pre-attenuated highly virulent parent SA14" for consideration at PLOS Pathogens. As with all papers reviewed by the journal, your manuscript was reviewed by members of the editorial board and by several independent reviewers. The reviewers appreciated the attention to an important topic. Based on the reviews, we are likely to accept this manuscript for publication, providing that you modify the manuscript according to the review recommendations.

Both reviewers commented that that the authors addressed a highly significant topic in the field of neurotropic viruses--ie, deciphering the mechanisms of neuroinvasion and neurovirulence. They appreciated the effort involved in making numerous recombinant viruses and using both cell culture and mouse models to identify viral determinants of JEV infection and disease. Although the present study establishes a solid foundation for additional mechanistic work in the future, I agree with the reviewer suggesting measurement of viral burden in neuronal and non-neuronal tissues of mice is necessary to address differences between neuroinvasion and neurovirulence. I also agree with the reviewer recommending modification of the manuscript text, particularly in the discussion and introduction, to highlight novel aspects of the present study--this is important, as several similar studies (but not to the same depth and breadth) as the present study have already been published.

Sincerely,

Sujan Shresta

Academic Editor

PLOS Pathogens

Alexander Gorbalenya

Section Editor

PLOS Pathogens

Michael Malim

Editor-in-Chief

PLOS Pathogens

orcid.org/0000-0002-7699-2064

Both reviewers commented that that the authors addressed a highly significant topic in the field of neurotropic viruses--ie, deciphering the mechanisms of neuroinvasion and neurovirulence. They appreciated the effort involved in making numerous recombinant viruses and using both cell culture and mouse models to identify viral determinants of JEV infection and disease. Although the present study establishes a solid foundation for additional mechanistic work in the future, I agree with the reviewer suggesting measurement of viral burden in neuronal and non-neuronal tissues of mice is necessary to address differences between neuroinvasion and neurovirulence. I also agree with the reviewer recommending modification of the manuscript text, particularly in the discussion and introduction, to highlight novel aspects of the present study--this is important, as several similar studies (but not to the same depth and breadth) as the present study have already been published.

Reviewer Comments (if any, and for reference):

Reviewer's Responses to Questions

**Part I - Summary**

Reviewer #1: The manuscript submitted by Song et al. (2024) to PLOS Pathogens seeks to elucidate the viral factors that lead to differences in pathogenicity (neuroinvasiveness and neurovirulence) between the attenuated Japanese Encephalitis Virus (JEV) vaccine strain SA14-14-2 and the virulent parental strain SA14. The authors identified mutational differences between the isogenic strains before utilizing a reverse genetic system to the generate full length infectious cDNAs of said strains (rSA14-14-2 and rSA14). Using the two cDNA’s as “genetic backbones”, the authors systemically swapped in different genomic regions between the two strains to generate chimeric genomes/viruses. Subsequently, the chimeric viruses were evaluated individually as well as compared to rSA14-14-2 and rSA14 in terms of their ability to replicate, produce viral proteins, and alter plaque morphology. The pathogenic capacity of the chimeric viruses was also evaluated in mice. Finally, the authors mapped the identified attenuating mutations onto the structures for envelope (E) protein, non-structural protein 1 (NS1), and non-structural protein 2A (NS2A). Throughout the manuscript, the authors do a thorough job of untangling and identifying the relative contributions of the non-coding regions, E, NS1/NS1’ and NS2A proteins on SA14 pathogenicity, ultimately providing strong evidence for the E protein serving as a key regulator of neurovirulence and neuroinvasion. This is not a novel conclusion (see Zheng et al. (2018) J Virol, Huang et al (2024) Virol J, Yu (2010) Vaccine), but is a detailed report on evaluating the contributions including and beyond the E protein. As such, this article would be suited to the pathogenesis and pathogen evolution audience of PLOS Path. Major and minor comments intended to improve the manuscript are outlined in more detail below.

Reviewer #2: Song et al. describe the construction of infectious clones of virulent parent and live attenuated vaccine strains of JEV. Several sets of chimeras are then created and analyzed for their growth properties in BHK-21 cells and their neuroinvasive and neurovirulent characteristics in CD-1 mice. Genetic determinants influencing these phenotypes are identified in E, NS1 and NS2A. The last section of the Results puts these determinants in the context of what is known about the functions of these proteins and their structures. The quality, scope and rigor of the molecular virology are outstanding, and the paper very well-written. One could argue that the paper falls short of uncovering the mechanisms by which these determinants influence neurovirulence and neuroinvasion (see below), but the work certainly sets the stage and provides a solid foundation for these kinds of future studies. The current submission represents a great deal of work, but I did find myself wondering about how the JEV parents and the chimeras would behave in an innate immunocompetent cell substrate. However, you could argue that this, again, could be a significant separate study.

The is one significant concern regarding neuroinvasion vs neurovirulence. The authors use 2 routes of infection (IM or IC) to interrogate wildtype, vaccine, and all chimeric strains in terms of neurovirulence and neuroinvasiveness. Survival is the only output presented. The results fully support the authors claims for the role of E - NS2A in neurovirulence. However, claims of neuroinvasiveness are neither supported nor unsupported by their data given that the vaccine strain and related derivatives lack lethality after intracranial challenge. It is therefore not possible distinguish between productive peripheral infection followed by non-neuroinvasion vs neuroinvasion followed by non-neurovirulence vs a lack of initial peripheral infection. It is possible that the vaccine strain and derivatives can invade the CNS but are rapidly cleared or the virus is eliminated from the periphery so rapidly that it precludes neuroinvasion.

**Part II – Major Issues: Key Experiments Required for Acceptance**

Reviewer #1: 1. Figure 2. It *seems* that the authors have mislabeled the run-off transcription site. Due to the mechanism and positioning of XbaI cleavage, and as the SP6 polymerase uses the reverse (“-“) strand as a template to generate the “+” strand, the run off site is actually tCTAG | not t | CTAG. As such, both genomes contain 4 extra residues (at least based on transcription). Please double-check this, but I think this is correct based on the orientation of your XbaI site as listed. It is unclear to me if you will get deletion of these extra nucleotides upon viral RNA replication since there is a strong bias for the “AG” dinucleotide for initiation of RNA synthesis on both strands in flaviviruses. I don’t think this changes the results described herein, but it is good to know that viral RNAs made with extra nucleotides are viable in cell culture. �

2. Figure 4: While presumably the choice is grounded in previous literature, it is not clear in the manuscript why the authors chose to neglect the potential influence on pathogenicity of other segments/genes beyond the ncrs and c-ns2a (adding a statement to this effect would be helpful). According to the sequencing data between the two strains (Figure 1), there seem to be 8 additional missense mutations scattered throughout ns2b, ns3, ns4b and ns5 gene segments. This is not the first study to report this (see reference 112, 114, 117 ect.). It would be helpful for the authors to comment on these in the discussion section.

3. Discussion, Paragraph 2, “E plays plays two key roles in JEV pathogenicity: It first acts as the primary initiator for neuroinvasion and subsequently as the sole determinant for neurovirulence”. It would be helpful to have more tie-in/speculation/discussion about how the identified attenuating mutations in E protein may affect entry (and subsequently pathogenicity).

4. Line 621/622 seems to be a bit of an overstatement based on the results from this study. While it is true that the NS2A protein appears to play a role in enhancing JEV neuroinvasiveness, the conclusion should read more as it does in the abstract, with NS2A touted to be a “[following E-mediated neuroinvasion, NS2A acts as a] secondary promoter, further amplifying viral neuroinvasiveness”.

5. In the last paragraph, there should be some reference or discussion surrounding the idea that NS2A has been previously associated with increased virulence and disruption of neurogenesis (see Ávila-Pérez et al. (2019) Sci. Rep and Yoon et al (2017) Cell Stem Cell).

Reviewer #2: To draw conclusions about neuroinvasion, additional assays such as viral titer and genomic RNA levels in the brain and periphery would be needed. Weight curves from the mice would also aid in monitoring disease progression in wild type and vaccine-challenged animals.

**Part III – Minor Issues: Editorial and Data Presentation Modifications**

Reviewer #1: 1. In the Introduction, the authors should consider defining enzymatic domains before jumping to abbreviations: protease/PRO (line 120), methyltransferase/MET (line 126), polymerase/POL (line 127), helicase/HEL (line 128)

2. In the Introduction, it would enhance reader understanding of NS1’ if there was more explanation provided to speak to its formation (line 122/123). IE “slippery heptanucleotide motif acts as a classic stimulatory motif for -1 ribosomal frameshifting with the downstream pseudoknot hypothesized to reinforce this action and promote a second, short reading frame extending into NS2A. This product is called NS1’”. You allude to these components in Figure 1, however without an explanation, the additional labelling may be unclear. Information comes from reference: Firth and Atkins (2009), Virol J

3. Figure 6/7: When stating the results of Figure 6/7, the authors fail to mention the potential impact that NS2A (with N2K and A40V) could be having on plaque size or pathogenicity as there is a no rSA14/NS2A or rSA14-14-2/NS2A. While this question is addressed in Figure 8, there should be some acknowledgement of the lack of this control in the text, but that it will be explored further in the next section.

4. Figure 8/9: The control, rSA14-14-2/e (which was present in the previous figure) is missing from Figure 8/9. Without this control, it makes it difficult to parse apart the contributions of E vs NS2A/prM/C. This is especially evident in Figure 9 whereupon without referring to Figure 6(D), the reader would have little concept of the neuroinvasive synergy that occurs when E and NS2A are replaced together (just the assumption that E alone is causing the changes).

5. Figure 8: To simplify the results for the reader, the authors should consider re-arranging Figure 8(D). In text, the authors draw the reader to different points of comparison between the plaque assay results - IE “rSA14-14-2/e+ns2a vs. rSA14-14-2/prm+e+ns2a and rSA14-14-2/c+e+ns2a vs. rSA14-14-413 2/c–e+ns2a”

Comparisons like this can be difficult to keep track of without visual groupings. It is suggested that either a) the totality of data is shown along with subgroupings of plaque results that the authors would like the reader to compare or b) reorganize the original panel to reflect the comparison groupings that the authors make (IE top down: rSA14-14-2/ ns2a, rSA14-14-2/e (see previous comment), rSA14-14-2/ e+ns2a, rSA14-14-2/ prm+e, rSA14-14-2/ prm+e+ns2a, rSA14-14-2/ c+e+ns2a, rSA14-14-2/ c−e+ns2

6. There were several aspects I appreciated about the manuscript and will briefly list a few here:

a. The generation of 20 chimeric viruses was commendable and gave weight to findings in both cell culture and mouse models.

b. It was appreciated that the authors separated out the effects of the viral proteins/ncrs on neuroinvasiveness versus neurovirulence

c. Figure 1A) was easy to digest, clean, and a great reference point for aspects mentioned later in the manuscript

Reviewer #2: None.

PLOS authors have the option to publish the peer review history of their article (what does this mean?). If published, this will include your full peer review and any attached files.

Reviewer #1: No

Reviewer #2: No

Figure Files:

Data Requirements:

Reproducibility:

References:

---

## [Decision Letter · Decision Letter 1]

17 Dec 2024

Dear Dr. Lee,

We are pleased to inform you that your manuscript 'Key virulence factors responsible for differences in pathogenicity between clinically proven live-attenuated Japanese encephalitis vaccine SA14-14-2 and its pre-attenuated highly virulent parent SA14' has been provisionally accepted for publication in PLOS Pathogens.

Best regards,

Sujan Shresta

Academic Editor

PLOS Pathogens

Alexander Gorbalenya

Section Editor

PLOS Pathogens

Sumita Bhaduri-McIntosh

Editor-in-Chief

PLOS Pathogens

orcid.org/0000-0003-2946-9497

Michael Malim

Editor-in-Chief

PLOS Pathogens

orcid.org/0000-0002-7699-2064

Reviewer Comments (if any, and for reference):

Reviewer's Responses to Questions

**Part I - Summary**

Reviewer #1: The changes submitted by Song et al. regarding the manuscript entitled “Key virulence

factors responsible for di4erences in pathogenicity between clinically proven live-

attenuated Japanese encephalitis vaccine SA14-14-2 and its preattenuated highly virulent

parent SA14” have satisfied the original reviewer comments. Major comments related to

relabeling/explanation of methods (major point #1), discussion points (major point #2,3,5),

and statement claims (major point 4) have all been addressed and altered throughout the

manuscript. The same rigor was applied to the minor comments, with all points taken

under advisement and changed or reworked accordingly. The introduction of Figure 10 with

associated data was appreciated and examined with no major comments to be added.

Reviewer #2: The authors have addressed my concerns.

**Part II – Major Issues: Key Experiments Required for Acceptance**

Reviewer #1: (No Response)

Reviewer #2: (No Response)

**Part III – Minor Issues: Editorial and Data Presentation Modifications**

Reviewer #1: (No Response)

Reviewer #2: (No Response)

PLOS authors have the option to publish the peer review history of their article (what does this mean?). If published, this will include your full peer review and any attached files.

Reviewer #1: No

Reviewer #2: No

---

## [Editor Report · Acceptance letter]

28 Dec 2024

Dear Dr. Lee,

We are delighted to inform you that your manuscript, "Key virulence factors responsible for differences in pathogenicity between clinically proven live-attenuated Japanese encephalitis vaccine SA14-14-2 and its pre-attenuated highly virulent parent SA14," has been formally accepted for publication in PLOS Pathogens.

Best regards,

Sumita Bhaduri-McIntosh

Editor-in-Chief

PLOS Pathogens

orcid.org/0000-0003-2946-9497

Michael Malim

Editor-in-Chief

PLOS Pathogens

orcid.org/0000-0002-7699-2064